# Multiverse Predictions for Habitability: Fraction of Planets that Develop Life

**McCullen Sandora** [1,2]

[1]   Institute of Cosmology, Department of Physics and Astronomy, Tufts University, Medford, MA 02155, USA; mccullen.sandora@gmail.com

[2]   Center for Particle Cosmology, Department of Physics and Astronomy, University of Pennsylvania, Philadelphia, PA 19104, USA

**Abstract:** In a multiverse context, determining the probability of being in our particular universe depends on estimating its overall habitability compared to other universes with different values of the fundamental constants. One of the most important factors in determining this is the fraction of planets that actually develop life, and how this depends on planetary conditions. Many proposed possibilities for this are incompatible with the multiverse: if the emergence of life depends on the lifetime of its host star, the size of the habitable planet, or the amount of material processed, the chances of being in our universe would be very low. If the emergence of life depends on the entropy absorbed by the planet, however, our position in this universe is very natural. Several proposed models for the subsequent development of life, including the hard step model and several planetary oxygenation models, are also shown to be incompatible with the multiverse. If any of these are observed to play a large role in determining the distribution of life throughout our universe, the multiverse hypothesis will be ruled out to high significance.

**Keywords:** multiverse; habitability; life

---

## 1. The Fraction of Habitable Planets that Develop Life

In this paper, we continue our investigation from References [1,2] into the probabilities of measuring our observed values of the fundamental physical constants $\alpha = e^2/4\pi$, $\beta = m_e/m_p$ and $\gamma = m_p/M_{pl}$ within a multiverse framework. Focusing on these three quantities supplements the traditional treatment of the cosmological parameters, and, as we will show, has the potential to go much further in terms of predictive power. The overarching goal of this investigation is to elevate the status of the multiverse to a traditional scientific theory, capable of making testable predictions that are verifiable on reasonable timescales.

The framework has been to use the principle of mediocrity [3,4], wherein the probability of measuring a given set of constants is directly proportional to the number of observers in universes with those constants. Since this is often a strong function of these physical constants, we expect to be in a universe that nearly optimally reflects what life needs. It is usually easier to tell what types of environments our universe is good at producing rather than determining the exact requirements for life, since the former relies only on physics, while the latter involves extrapolating from biology. This approach relies on a notion of habitability: that is, the precise conditions required for the emergence of observers, here identified with life that is 'complex enough'. Herein lies the predictive nature of this enterprise: as there is no strong consensus on the conditions for life to arise and survive long enough to develop intelligence, we investigate a multitude of possibilities. We then tabulate which are compatible with our existence in this universe, and which would imply the existence of much more fecund universes that host the majority of observers. In the latter case, our universe would be a backwater, so

much so that the probability of being one of those few observers here is low, sometimes to an extreme degree. This provides a test of these notions of habitability themselves: once we accrue large amounts of exoatmoshperic measurements from a diverse array of planetary environments, which is poised to happen in the coming decades, we will be able to correlate which, if any, are capable of hosting life. This will allow us to finally determine whether the true habitability condition matches the multiverse predictions. Because there are dozens of potentially important planetary and stellar characteristics, and the multiverse would be falsified if just one of them were deemed to be incompatible, this will serve as a very efficient method for putting this overarching framework to a rigorous test.

To estimate the number of observers in the universe, we use the Drake equation [5], which factors this question into subcomponents that are more or less capable of functioning in isolation. If this prescription is taken, then the probability of observing our values of the physical constants is

$$
\begin{aligned}
\mathrm{P}(\alpha, \beta, \gamma) \quad &\propto \quad p_{\mathrm{prior}}(\alpha, \beta, \gamma)\, N_\star(\alpha, \beta, \gamma) \int d\lambda \, p_{\mathrm{IMF}}(\lambda, \alpha, \beta, \gamma)\, h(\lambda, \alpha, \beta, \gamma) \\
h \quad &= \quad f_{\mathrm{p}} \times n_{\mathrm{e}} \times f_{\mathrm{bio}} \times f_{\mathrm{int}} \times N_{\mathrm{obs}}
\end{aligned}
\tag{1}
$$

Here, $N_\star$ is the number of stars in the universe, $p_{\mathrm{IMF}}$ is the initial mass function, and $h$ defines the notion of habitability of a given environment, defined as the likelihood that it gives rise to the emergence of observers. It may be worth bearing in mind that this differs from the definition that astrobiologists typically use, who are more focused on the occurrence of unicellular life. This quantity naturally factorizes into a product of separate factors, which are: the fraction of stars that have planets $f_{\mathrm{p}}$, the number of habitable planets per planet-hosting star $n_{\mathrm{e}}$, the fraction of planets that develop life $f_{\mathrm{bio}}$, the fraction of life-bearing worlds that develop intelligence $f_{\mathrm{int}}$, and the total number of observers on intelligence-bearing worlds $N_{\mathrm{obs}}$. In addition the factor $p_{\mathrm{prior}} \sim 1/(\beta\gamma)$ is used to account for the relative frequency of occurrence of each type of universe, derived from high energy physics considerations. All of these quantities may depend on the fundamental constants, as well as the local environment, which in our analysis is restricted to the stellar mass $\lambda$ (made dimensionless by comparing to the natural scale $(8\pi)^{3/2} M_{pl}^3/m_p^2$). Our strategy has been to estimate the overall probability by working our way through these factors from left to right, incorporating our previous findings for various habitability proposals into a cohesive analysis.

In Reference [1] we considered the number of habitable stars in a universe as a function of our physical constants, for various definitions of habitability. The main stumbling block is the strength of gravity, which is capable of being 2 orders of magnitude larger without affecting habitability in any obviously adverse way. If gravity were stronger, stars would be smaller, and so there would be more of them, and if each represents an independent opportunity for life to evolve, there would be more observers in those universes. We also incorporated various other potential habitability criteria, such as the requirement that planets not be tidally locked, that stars be not fully convective, that starlight be photosynthetic, and that stars last for biological timescales. While none of these were capable of rescuing the multiverse hypothesis, the tidally locked and photosynthesis conditions will be crucial components of our discussions here.

This was extended to the study of planets in Reference [2]. There are two separate terms related to this in the Drake equation, the first of which being the fraction of stars with planets. While recent results indicate that a minimum metallicity is required for protoplanetary disks to form planets, the constraints on parameters from this condition are quite mild, indicating that planets themselves are generic features throughout a range of alternate universes.

The second planet-related factor is the average number of habitable planets found around stars that do possess them. Here, we address various planetary habitability criteria. It is pointed out that our universe seems to preferentially produce roughly Earth mass planets, out of the eight orders of magnitude it could have chosen. If we take the assumption that Earthlike planets are necessary to host life, this fits in quite well with the multiverse hypothesis, almost regardless of the specific planet formation scenario one employs. Also of note is that the width of the temperate zone is roughly

equivalent to the interplanetary spacing (around sunlike stars), ensuring that some planet will be capable of supporting liquid water on its surface in essentially every planetary system.

In this work, we extend our previous analysis to the fraction of planets that develop simple life. This is of course unknown at the moment, and many of the guesses we make about this will be utterly incompatible with the multiverse hypothesis. However, we demonstrate one that is fully capable of making our values typical, and discuss the associated distribution of life that we expect based off this. We first consider that the emergence of life depends on the amount of time, the planetary size, and the entropy production of the host star, and find that only the last is compatible with the multiverse, and further that it is fully able to bring the predictions into alignment with observation. In other words, we have discovered what our universe is truly good at: producing lots of entropy.

An important aspect of our analysis in this paper is not just the probability of observing our constants, but also our particular position within our universe. Many environmental parameters should be assessed, including planetary mass, orbit, metallicity, amount of water, carbon to oxygen ratio, and so forth, but we restrict our analysis to stellar size here. This becomes especially important when considering that habitability is proportional to entropy production: taken blindly, this leaves unanswered why we do not live around a much smaller star at a much later point in the future. This may be used to argue that the cutoff mass for stellar habitability is not too much below the solar value. This consideration favors several of our previously proposed habitability criteria, including the tidal locking and yellow light conditions. Synthesizing this with the analysis we perform on the physical constants is able to more powerfully constrain the viable habitability criteria.

After this, we consider a few more geologically inspired conditions: the notion that the size of the biosphere is limited by the amount of nutrient flux on a planet is investigated, and shown to fare worse than the entropy limited case. The question of whether radiogenic plate tectonics is necessary for life is addressed, and this condition is shown to also do worse than the case where it is ignored, though in some cases not terribly so.

We extend our analysis to more sophisticated accounts of the emergence of complex life to determine which of these are compatible with the multiverse. Chiefly, we examine the hard step model, the bated breath model, and the easy stroll model. We find reason to strongly disfavor the hard step model, both within the multiverse context and also on a biological basis. Multiple scenarios for atmospheric oxygenation are investigated, all sufficiently explaining our appearance toward the end of our planet's habitable phase within our universe, but all failing to explain this coincidence in the multiverse context. We determine that only the last is fully compatible with the multiverse.

## 2. What Factors Influence the Emergence of Life?

As explicitly incorporated into the Drake equation, we specialize our discussion of the emergence of life to planets orbiting stars. We now wish to estimate the fraction of habitable planets which develop life. This is likely to depend, at least to some extent, on the properties of the planet under consideration, and the distinction between factors is not always as clear cut as it may first appear. For instance, in Reference [2] we investigated different notions of the definition of habitability, such as the size and temperature of a planet, which could just as easily have been classified as affecting $f_{bio}$. Here we specify to temperate, terrestrial planets, and ask what may further influence the emergence of primitive, that is, microscopic, life. There are a number of conceivable factors that may influence this rate: the time in the temperate zone, planetary size, the amount of entropy and nutrients processed, the presence or absence of plate tectonics, and so forth. These will in turn be considered here, but this is far from an exhaustive list. We will succeed in showing that many of the reasonable expectations for the factors dictating where life can emerge will turn out to be incompatible with the multiverse hypothesis. In turn, this will lead us to some definite predictions for where life should be found in our universe.

*2.1. Is Habitability Proportional to Stellar Lifetime?*

As a first trial, we assign a habitability that is linearly proportional to the lifetime of the star, $h(\lambda) \propto t_\star(\lambda)$. The reasoning behind this is that if life typically takes very long to develop, then the chance of it arising around any given star will be small, but will grow with the star's total lifetime. This stands in contrast to our naive treatment in Reference [1], where we treated all stars as equihabitable, $h(\lambda) \propto 1$. (We also crudely accommodated stellar lifetime in this setup by optionally considering $h$ to be a simple step function of the lifetime of the star: here, a star was deemed habitable if its lifespan exceeded a certain number of 'ticks of the molecular clock', which we took to be $N_{\rm bio} \sim 10^{30}$ to equate to several billion years, and uninhabitable otherwise. This allowed us to place an absolute upper limit on the strength of gravity, $\gamma < 134\gamma_{\rm obs}$, as above this value no star would have a suitable lifetime.) Let us also note that a more general time dependence, along the lines of Reference [6], may be expected: our analysis in this section can represent a first order Taylor expansion, valid when probabilities are always small. Generalizations to this are considered below.

We use that the stellar lifetime is $t_\star(\lambda) = 110\alpha^2 M_{pl}^2/(m_e^2 m_p \lambda^{5/2}) \equiv \hat{t}_\star/\lambda^{5/2}$ [7]. To make a comparison to other universes, this needs to be divided by another timescale to define a dimensionless ratio: here we use the molecular timescale given by the expression $t_{\rm mol} = 27 m_p^{1/2}/(\alpha^2 m_e^{3/2})$. This ratio counts the total number of interactions any given molecule experiences throughout the star's lifetime. Then, using the fact that $\int d\lambda \, p_{\rm IMF}(\lambda) \lambda^q \sim \lambda_{\rm min}^q$ because $\lambda_{\rm min}$ is the only scale in the initial mass function, we arrive at

$$\mathrm{P}_{t_\star} \propto p_{\rm prior} \, N_\star \, \frac{\hat{t}_\star}{N_{\rm bio} \, t_{\rm mol}} \frac{1}{\lambda_{\rm min}^{5/2}} \propto \frac{\beta^{9/8}}{\alpha^{5/4}} \tag{2}$$

One interesting aspect of this expression is that the dependence on $\gamma$ entirely drops out. This indicates that the total habitable time in a universe is independent of this quantity, as although there are more stars in universes with stronger gravity, they last longer in universes with weaker gravity, and there are more of these universes to exactly compensate any preference. What does change is the number of stars this total time is divided among, but because of the simple linear relationship, life would be indifferent to this partitioning. Clearly, this indifference must break down at extreme values of this parameter, that would either create a small number of nearly indefinite stars, or else a cornucopia of exceedingly briefly shining objects. However, with this criterion we would not expect to be situated as we are, more than two orders of magnitude away from the upper boundary used in Reference [1]. The distribution of observers throughout the multiverse is plotted in Figure 1. The probabilities for this habitability criterion, defined as the smaller of P and $1 - \mathrm{P}$, are[1]:

$$\mathbb{P}(\alpha_{obs}) = 0.251, \quad \mathbb{P}(\beta_{obs}) = 0.196, \quad \mathbb{P}(\gamma_{obs}) = 0.007 \tag{3}$$

These can be compared to the values $\mathbb{P}(\alpha_{obs}) = 0.20$, $\mathbb{P}(\beta_{obs}) = 0.44$, and $\mathbb{P}(\gamma_{obs}) = 4.2 \times 10^{-7}$ that were found by simply taking $h(\lambda) = 1$. Though the probability for observing our $\gamma$ in particular is orders of magnitude better than what we had found without weighting by stellar lifetime, it is still disquietingly small. We conclude that habitability can not be a simple linear function of stellar lifetime, otherwise we would be in a universe where gravity was stronger.

This conclusion has an important corollary: if habitability cannot depend on stellar lifetime, we can conclude that older stars should not be more likely to host biospheres. This expectation makes explicit use of the multiverse hypothesis, and so if a future catalog of biospheres displays a correlation with stellar age, it will constitute evidence against the multiverse, at the level of $2.7\sigma$.

---

[1]    The code to compute all probabilities discussed in the text is made available at https://github.com/mccsandora/Multiverse-Habitability-Handler.

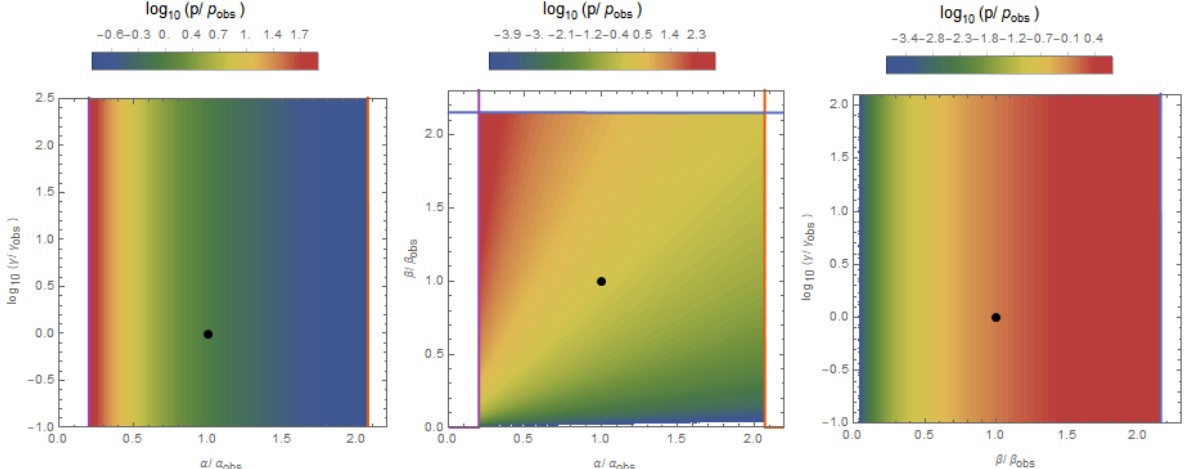

**Figure 1.** Distribution of observers from imposing the stellar lifetime condition. The black dot denotes the values in our universe, and the orange, blue, and purple lines are the hydrogen stability, stellar fusion, and galactic cooling thresholds, respectively, discussed in Reference [1].

### 2.2. Does Habitability Depend on Planetary Size?

The previous treatment of habitability was indifferent to the aspects of the planet in question. In general the habitability will depend on a great variety of factors, including size, mineral and volatile composition, amount of atmosphere and ocean, irradiance, source of internal heat, spin, orbit, obliquity, eccentricity, presence of any moons, possible secular resonances, and overall solar system architecture. Here we neglect all of these potentially important factors aside from size: the others constitute habitability hypotheses in their own right which can be incorporated into this analysis in the future. For the moment we restrict our attention to terrestrial worlds: that is, worlds capable of retaining a marginal atmosphere. In Reference [2] we discussed how this selects a relatively narrow range of planetary masses, characterized in terms of physical constants as $R_{\text{terr}} = 3.6 M_{pl}/(\alpha^{1/2} m_e^{3/4} m_p^{5/4})$. For the present purposes we assert that the fraction of stars that will have a planet of this size as independent of both the underlying parameters and stellar mass (evidence supporting this assumption can be found in, for example, Reference [8]). This will be paired with the more sophisticated treatment we undertook in Reference [2] in Section 4, but this will not affect our qualitative results.

Changing the physical constants will change the size of terrestrial planets. It is not difficult to imagine that the larger a planet, the greater the chances life will arise, essentially because of the greater number of experiments its chemical soup would be able to carry out [9]. This is bolstered by the observation that Earth, the one planet we know of that possesses life, is the largest terrestrial planet in our solar system [10]. For this, we define the habitability of a planet to be $h \propto N_{\text{interactions}} = N_{\text{sites}} t_\star/t_{\text{mol}}$, the number of chemical interactions that occur over the planet's lifetime. This weights the previous estimates based solely off lifetime by the number of active sites a planet contains.

Of course, this is a highly simplistic method of taking size into account. Much work has been done on what are termed superhabitable worlds recently [10], which asks the question of how the habitability properties may scale with, among other things, planetary size. There it was pointed out that larger worlds may very well be less habitable, because plate tectonics may not be operational, or because continents may be larger, yielding proportionally more desert regions, and so forth. Likewise, smaller planets may be expected to be less habitable because they cannot retain their atmospheres, cool more quickly, and may not possess a protective magnetic field. How planetary properties scale with size in our universe is a different question than how they scale with values of fundamental parameters, however, though the one may potentially inform the other.

The number of sites will not scale as simply as $(R_{\text{terr}}/L_{\text{mol}})^2$, however; a more nuanced analysis must be carried out. To estimate the total number of reaction sites we follow Reference [11], where the number of sites is estimated as

$$N_{\text{sites}} \sim \frac{V_{\text{clay}} \rho_A}{L_{\text{mol}}^2} \sim \alpha^{3/2} \beta^{3/4} \gamma^{-3} \tag{4}$$

Here, several quantities were used: the total amount of clay upon which chemical reactions can take place is given roughly by the average depth of clay times the surface area of the Earth. This depth is set by the same physics that yields the size of mountains, as we detail in the Appendix: it is set by equating the gravitational energy to the molecular energy, though the average depth of clay is several orders of magnitude smaller than a typical mountain, on account of the chemical bonds being much weaker. Then we have $H_{\text{clay}} \sim 0.01 H_{\text{mountain}} \sim 0.01 E_{\text{mol}}/(g m_p)$, and $V_{\text{clay}} \sim 4\pi R_{\text{terr}}^2 H_{\text{clay}}$. Note that the height of mountains scales inversely with the planetary radius, so that the number of sites is actually linear in radius. We also need the 'surface area per volume' $\rho_A$ of typical clay, which takes into account the high fractal dimension of the mineral surface: in Reference [11] this was estimated to be $10^{-6} \text{cm}^{-1}$, which in terms of physical constants we take to be set by the size of molecules, given by the Bohr radius.

Taking this hypothesis yields

$$P_{\text{size}} \propto \frac{\alpha^{1/4} \beta^{15/8}}{\gamma^3} \tag{5}$$

The distribution of observers for this is plotted in Figure 2. This gives the probabilities

$$\mathbb{P}(\alpha_{obs}) = 0.37, \quad \mathbb{P}(\beta_{obs}) = 0.11, \quad \mathbb{P}(\gamma_{obs}) = 0.01 \tag{6}$$

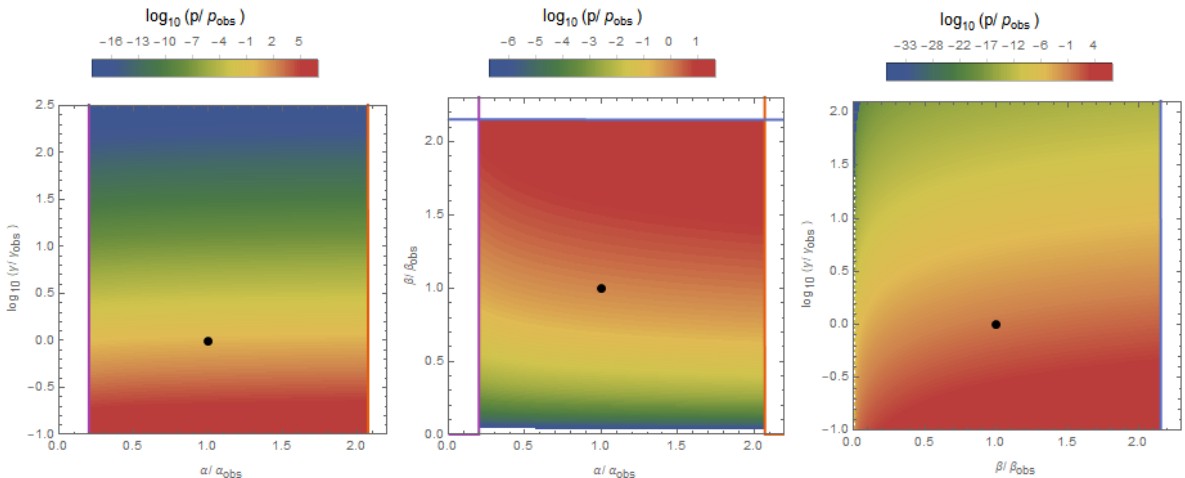

**Figure 2.** Distribution of observers from imposing the size condition.

This notion that the habitability of a planet should scale with its size is in conflict with what we observe, because our universe does not favor particularly large habitable planets. Here the main conflict is again due to the strength of gravity, though the dependence that plagued the criteria of our previous analyses has now been inverted, so that now extremely small values are preferred. The estimate we presented was taken using $\gamma_{\text{min}} = 0.1\gamma_{\text{obs}}$, though we do not find any lower bounds on this quantity from any of our considerations. From this, we find that if planet size does dictate habitability, the multiverse hypothesis will have made a wrong prediction.

However, including some of the criteria we have discussed in previous papers can ameliorate this situation. Insisting that tidally locked planets are uninhabitable raises the probability for observing our value of $\gamma$, giving

$$\mathbb{P}(\alpha_{obs}) = 0.10, \quad \mathbb{P}(\beta_{obs}) = 0.09, \quad \mathbb{P}(\gamma_{obs}) = 0.06 \tag{7}$$

This is because for small $\gamma$, the dependence is tempered by a factor of $(\lambda_{\min}/\lambda_{\mathrm{TL}})^{4.85}$. This makes the probability proportional to $\gamma^{-1.24}$, alleviating the strong preference for smaller values. Similar results hold if the photosynthesis condition is applied as well. However, the increase in the probability of $\gamma$ here is compensated by a decrease in the probability of $\alpha$.

Then, the size condition may be kept with certain caveats: the size of a planet may in fact be important, but only if tidally locked planets are uninhabitable and/or photosynthesis is required. It is certainly not as clean as being able to discard this notion of habitability in its entirety, but it illustrates the state of affairs we hope to achieve. Of the laundry list of potential conditions necessary for life, only certain combinations will be compatible with the multiverse hypothesis, and if complicated conditionals must be employed in order to check consistency, then so be it.

### 2.3. Is Habitability Dependent on Entropy Production?

One further quantity that the development of complex life may depend on is the entropy produced on the planet per unit time, which serves as an upper limit for the rate of information processing a biosphere can hypothetically manage. On Earth, entropy production is dominated by the downconversion of sunlight to lower frequencies, which yields approximately

$$\dot{S} \sim \frac{L_\star}{T_\star} \frac{R_{\mathrm{terr}}^2}{4a_{\mathrm{temp}}^2} \sim 10^{-3} \frac{\alpha^{13/2}\,\beta^4}{\lambda^{19/40}\,\gamma^{9/4}} \sim 10^{36} \frac{\mathrm{bits}}{\mathrm{sec}} \tag{8}$$

We have made use of the estimates for all these quantities from the appendix of Reference [1], which are stellar luminosity $L_\star$, stellar surface temperature $T_\star$, and temperate orbit $a_{\mathrm{temp}}$. Note that here, we have specified to planets that orbit within the temperate zone, at which liquid water can exist on the surface. We also assume that stellar temperature is much greater than that of the planet, which holds for all main sequence stars; to extend this analysis to systems such as brown dwarfs, refinements such as found in Reference [12] should be used.

This can be compared to estimates for the total information processed by the biosphere, which was estimated as $\dot{S}_{\mathrm{biosphere}} = 10^{39}$ bits/sec in Reference [13]. The fact that this is higher than the incident entropy production is not an indication of the violation of the second law of thermodynamics, as the authors admit that their figure is likely to be an overestimate, based off of rates measured in metabolically active bacteria cultured in the lab. If we try ourselves by using the entropy of a single bacterium $S_{\mathrm{bact}} = 2 \times 10^{11}$ from Reference [14] and the cell turnover rate of $1.7 \times 10^{30}$ cells/year from Reference [15], we find $\dot{S}_{\mathrm{biosphere}} = 10^{34}$ bits/s, which is 1% of the total information processing available. This is in line with the result that biological information processing systems universally converge to several orders of magnitude below the theoretical limit [16], the rest being converted into waste heat. What is of note, however, is that these two numbers are indeed comparable, signaling that the ultimate size of the biosphere [17] (and ultimately technosphere [18]) is foremost limited by the amount of possible information that can be processed in its environment. If the emergence of life were dependent on the amount of information processed, rather than the number of ticks of the molecular clock, we would expect this quantity to be selected for.

The entropy production rate can also be used to determine the size of the biosphere by considering the amount of entropy produced per molecular time, $\Delta S \sim \dot{S} t_{\mathrm{mol}}$. This was considered in References [19] and [20], where it was shown that the requirement that planets be large enough to host biospheres of sufficient complexity to contain conscious societies did not serve as a very strong constraint on physical parameters. This constraint will not be considered further here.

More appropriately, we may take the presence of complexity to be dependent on the total amount of entropy delivered to the system (as before, this obviously breaks down in its extreme limits, such as if all the entropy were delivered within a single minute). This is actually a more natural choice than just considering the amount of time a planet spends in the habitable zone, as the rate of evolution should be weighted by the overall size of the system doing the exploration [21]. In this case, we have

$$\Delta S_{\text{tot}} \sim \dot{S}\, t_\star \sim \frac{\alpha^{17/2}\,\beta^2}{\lambda^{119/40}\,\gamma^{17/4}} \sim 10^{54} \tag{9}$$

Let us also note that we have been purposefully vague as to what we are trying to encapsulate with this criterion: how can the probability of the emergence of life depend on the size of biosphere? This is a major presupposition. Rather, what we are actually computing represents the probability that a given biosphere can attain some given state, be that intelligent observers, multicellularity or whatever else. As such, this may more naturally be classified under one of the other Drake factors, such as $f_{\text{int}}$. Our unwillingness to commit to a definite interpretation of this quantity justifies including it in the current discussion instead.

Before using this to estimate probabilities, an important caveat must be made: the total entropy itself should not be important unless it can be utilized by living organisms. This is achieved on our planet through the process of photosynthesis, whereby sunlight is converted into chemical energy. The size of the biosphere must be conditioned on the fact that the star's light be within the chemically absorptive range, a feature that was discussed originally in Reference [22] and at length in Reference [1]. Due to this fact, the estimate for the probability of observing certain values of the constants does not attain as simple a form as our estimates above, but nevertheless can be computed,

$$\mathbb{P}_{\text{S}} \propto \frac{\alpha^{2.54}\,\beta^{3.98}}{\gamma^{2.25}} \left( \min\left\{1, 0.45\frac{L_{\text{fizzle}}}{1100\text{ nm}}Y^{1/4}\right\}^{9.11} - \min\left\{1, 0.16\frac{L_{\text{fry}}}{400\text{ nm}}Y^{1/4}\right\}^{9.11} \right) \tag{10}$$

Here, $Y = 3.19\alpha^{-63/20}\beta^{137/40}\gamma$ and the length scales that appear delimit the wavelengths of photosynthetic light. Here, we take the optimistic upper bound taken from Reference [23] on the basis that the light be above the thermal background and the lower bound from Reference [24] to avoid photodissociation. The distribution of observers for this criterion is displayed in Figure 3.

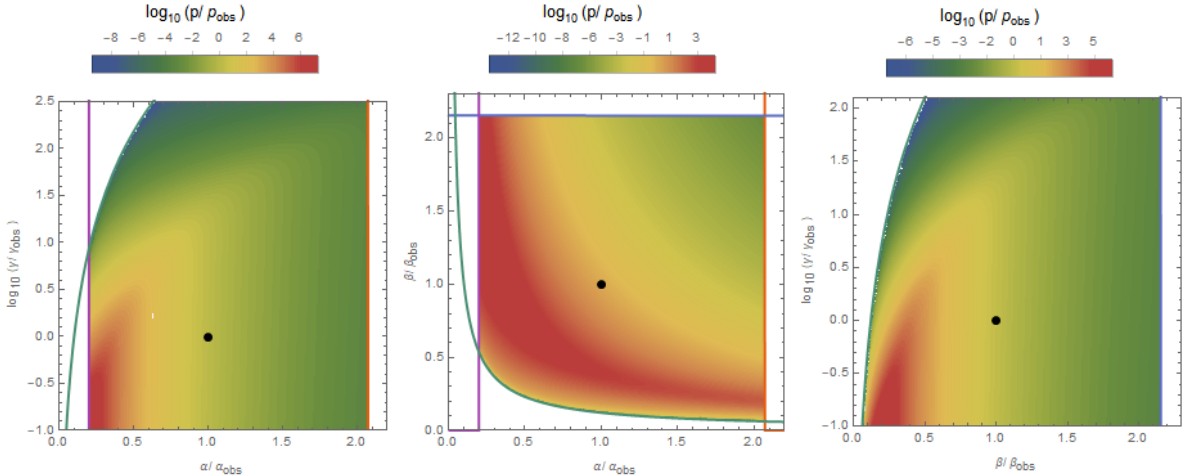

**Figure 3.** Distribution of observers from imposing the entropy condition.

For optimistic values of the potential photosynthetic range, $400\text{ nm} < L < 1100\text{ nm}$, the corresponding probabilities are

$$\mathbb{P}(\alpha_{obs}) = 0.24, \quad \mathbb{P}(\beta_{obs}) = 0.38, \quad \mathbb{P}(\gamma_{obs}) = 0.38 \tag{11}$$

This will be referred to as the 'photosynthesis condition'. For more pessimistic values of the photosynthetic range, 600 nm $< L <$ 750 nm, which we refer to as the 'yellow condition', we have

$$\mathbb{P}(\alpha_{obs}) = 0.19, \quad \mathbb{P}(\beta_{obs}) = 0.44, \quad \mathbb{P}(\gamma_{obs}) = 0.32 \tag{12}$$

Suffice it to say, this habitability criteria is fully consistent with the multiverse hypothesis, and not very sensitive to the photosynthetic range used. It has implications for the distribution of observers that may be eventually tested: we should expect to find complex life in those locales with the most amount of entropy production. While fully determining the places this distinguishes will rely on an in-depth analysis, this would include planets that orbit more active stars, for longer, and able to collect more incident radiation. This last criterion would include planets which orbit closer to their host star and are perhaps as large as can be, within the ranges compatible with life.

These predictions bring a certain amount of subtlety, however: at first glance they seem to be in direct conflict with the results of the previous two sections, that we should expect no correlation of life with stellar lifetime or planetary size. The distinction here is that life's presence should only depend on these quantities inasmuch as they determine the entropy collected. This is not degenerate with the criteria of before, though the number of samples needed to distinguish these two scenarios is left for future work.

However well this criterion may do in explaining the observed values of our constants, it fails to account for our location within our universe on its own. This is an equally powerful test of which habitability criteria are compatible with observations, and so we now turn our attention to this as well.

### 2.4. Why Are We Around a Yellow Star?

With the inclusion of the entropy production criteria, we have a notion of habitability that makes our observed values of the three microphysical parameters we focused on consistent with the multiverse model. Now, it is necessary to include local observables to further test the consistency of this criterion: namely, if the probability of life arising around a star is proportional to the total amount of entropy it produces over its entire lifetime, we must ensure that this is compatible with our presence around a star such as our sun. This consideration is capable of yielding extra information about where we should expect life to be in our universe: since smaller stars produce significantly more entropy over their lifetimes, we should expect some cutoff not too far below 1 solar mass where complex life cannot develop.

Restricting our attention to within our universe, then, we may ask what the probability is that we find ourselves around a star of one solar mass. This has been the subject of recent investigation, for instance in References [25–27]. Of course, this places us in the undesirable situation of trying to conduct a statistical analysis based off of a sample size of one, and with heavy selection effects at that. A more robust question would be to predict the distribution of biospheres as a function of stellar mass, which can test the model more concretely. Until we are technologically able to perform such measurements, however, we focus on the immediately accessible question. For a generic definition of the habitability of a system, the probability of being around a solar mass star or larger is

$$\mathrm{P}(M_\odot) = \frac{\int_{\lambda_\odot}^{\infty} d\lambda \, p_{\mathrm{IMF}}(\lambda) h(\lambda)}{\int_{\lambda_{\min}}^{\infty} d\lambda \, p_{\mathrm{IMF}}(\lambda) h(\lambda)} \tag{13}$$

For the simplest habitability hypothesis that all stars are equally habitable, we find that $P(M_\odot) = 0.14$, since approximately 14% of stars are larger than the sun (in agreement with contemporary surveys and Reference [25]). This is a perfectly reasonable account for our current position within our universe. However, we remind the reader that it failed miserably at accounting for the values of the constants themselves. If we instead use the entropy condition, the probability of being around a smaller star is weighted much higher: $h(\lambda) \sim \lambda^{-3}$. This is a direct consequence of the lower temperature and especially the longer lifetime of small mass stars. If this habitability hypothesis is used, we instead find that $P(M_\odot) = 0.02$, so that only 1 in 50 civilizations should expect to be around a star this large

(not to mention this early [26]). Thus, neither habitability hypothesis can simultaneously explain our position in our universe and within the multiverse itself.

However, these are not the only two notions of habitability we have encountered- far from it. If we include the 4 new possibilities along with the 480 from References [1,2], this brings the total to 1920 separate habitability criteria to test. Since the aim is now to explain why we do not live around a smaller star as well as why we live in this universe, we focus on those criteria that penalize small mass stars. From before, we had three of these: considering tidally locked planets to be uninhabitable rules out stars below $0.85 M_\odot$, if convective stars are uninhabitable the minimum is $0.35 M_\odot$, and if only yellow light can be photosynthetic the minimum is also $0.85 M_\odot$. The probabilities for each of these are displayed in Table 1. Of these potential explanations, the convective criterion does nothing to alleviate the problem, since the cutoff is below even the most optimistic photosynthetic mass. The other two hypotheses are understandably similar, since they introduce the same low mass cutoff. They are not identical because the yellow criterion also introduces a high mass cutoff at $1.3 M_\odot$, but both of these work even better than the equihabitable criterion.

**Table 1.** Probability of orbiting a star larger than or equal to our sun with the various habitability hypotheses. Whenever the entropy condition is used, the photosynthesis condition is also employed, except for the 'none' and 'yellow' rows. Since the size and nutrient flux conditions have the same dependence on $\lambda$ as the stellar lifetime condition, they all have the same probability of being around the sun.

| Criteria | $P(M_\odot)$, $h \propto 1$ | $P(M_\odot)$, $h \propto S$ | $P(M_\odot)$, $h \propto t_\star$ |
|---|---|---|---|
| none | 0.142 | $1.3 \times 10^{-4}$ | $4.3 \times 10^{-4}$ |
| TL | 0.835 | 0.528 | 0.570 |
| convective | 0.345 | 0.024 | 0.030 |
| photo | 0.308 | 0.024 | 0.038 |
| yellow | 0.585 | 0.424 | 0.449 |

When we considered the effects each of these criteria on the multiverse probabilities in Reference [1], we found no strong preference for whether to expect stars of these sorts to be habitable. Now that we incorporate additional criteria, however, they become crucial. This is due to the fact that because we place a strong preference on high entropy production, this favors stars that produce more than our sun. We need some sort of reason, then, why low mass stars are inhospitable. While the presence of convective flares, tidal locking, or absence of photosynthetic radiation are all reasonable hypotheses, only the latter two are coherent explanations. While we cannot uniquely specify the reason for the inhospitability of low mass stars, we end up with the prediction that either life cannot thrive on tidally locked planets or that photosynthesis is only possible with yellow light (or both). Flare stars may be uninhabitable too, but this does not constitute as good an explanation of our star's mass as it first appeared to.

We also explore the possibility that while a higher entropy production will be more conducive to the development of life, at some point the dependence must turn over, as the probability of development saturates to a near certainty. This may be encapsulated in the trial function $h(\lambda) = 1 - e^{-\Delta S(\lambda)/S_0}$. If $S_0$ is large compared to all produced stellar entropies considered, this recovers the analysis from before, whereas if $S_0$ is small the probability is essentially 1. Intermediate values interpolate between these extremes. One may think that if the value of $S_0$ is close to the solar value for whatever reason, this may naturally explain our presence in this universe without the need to invoke a large value for the smallest habitable star. The probabilities of our constants, as well as of being around a sunlike star, are plotted in Figure 4 as a function of $S_0$, where we find the interpolating behavior as advertised: for small $S_0$, it tends toward the photosynthesis criterion, which has the probabilities $\mathbb{P}(\alpha_{obs}) = 0.44$, $\mathbb{P}(\beta_{obs}) = 0.18$, and $\mathbb{P}(\gamma_{obs}) = 8.4 \times 10^{-7}$, whereas for large $S_0$ it tends toward the entropy condition. Intermediate values fail to simultaneously account for $P(M_\odot)$ and $P(\gamma_{\text{obs}})$, with one of these quantities being below 6% for any choice of $S_0$. So, while there may very

well be some amount of entropy production that almost guarantees that life will arise, there is no reason to expect that it is anywhere close to the amount so far produced by the sun, and it plays no role in explaining the mass of our star.

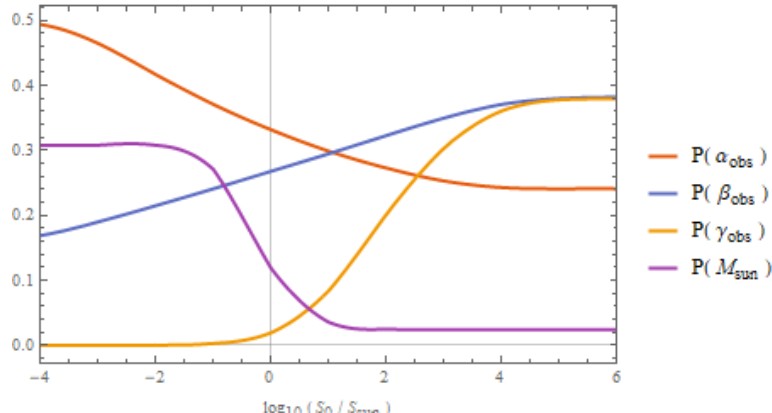

**Figure 4.** Probabilities with differing interpolation entropy, assuming the entropy production and photosynthesis criteria. This interpolates between the photosynthesis criterion for small $S_0$ and linear scaling for large $S_0$.

*2.5. Is the Biosphere Entropy or Material Limited?*

Above, we have shown that treating the biosphere as set by the total amount of entropy produced yielded the best account for our position within this universe, and have quoted several studies suggesting that this may indeed place the ultimate limit on biosphere size. However, there is plenty of reason to be skeptical of this claim: often ecosystems are instead resource limited on Earth [28], and are expected to be elsewhere as well [29]. Much of the discussion on the total primary productivity is centered on exactly which nutrient is the limiting factor for growth [30].

This being said, there are a few indications that it is indeed entropy that sets the ultimate limit of the size of the biosphere. While if one nutrient is found to be scarce it can be recycled many times, there is no real method for recycling light energy. Indeed, phosphorus, which is often a limiting factor, can be recycled as many as 500 times before leaving the biosphere [31]. Plankton have the ability to substitute many of the trace metals for each other in their various enzymes to take advantage of any local imbalance, and it has been found that the availabilities of each nutrient are roughly equal [32]. This colimitation is a natural outcome of life adjusting its activity to bolster its utilization of one resource, until the point where such optimization would no longer be beneficial. The fact that one of the colimiting factors can be light has been demonstrated to occur in subarctic ecosystems, where a concomitant increase in irradiance and iron flux lead to the largest amount of phytoplankton growth [33]. This indicates that though the tropics have more than enough light energy to sustain the same level of productivity year round, the balance between nutrients and light are roughly comparable. Thus, the biosphere seems to push recycling capacity until it hits the hard limit, dictated by the total amount of entropy that can be harnessed.

That being said, here we adopt the traditional stance that the size of the biosphere is limited by nutrient flux. In this scenario, the total mass of living organisms is set by the rate at which material that is weathered. The details of how to estimate this are relegated to the Appendix, but the result is

$$\Delta C_{\text{tot}} \sim 10^3 \, \epsilon_C \, \frac{\alpha^{9/2} \beta^{1/2}}{\lambda^{5/2} \gamma^3} \tag{14}$$

The biosphere size that can be supported depends on the actual residence times of each nutrient, which depend on geochemical and hydrological factors that we do not attempt to model here. Instead, we must content ourselves with parameterizing this in the efficiency factor $\epsilon_C$ for the time being,

trusting that the overall scaling will not be altered by too much. The scalings in this quantity are not too different from the entropy limited case in Equation (9). However, this criterion does not perform as well: if we impose that photosynthesis is necessary in this case as well, to facilitate comparison (as well as ensuring that the probability of orbiting a sunlike star is not too low), the probabilities are

$$\mathbb{P}(\alpha_{obs}) = 0.09, \quad \mathbb{P}(\beta_{obs}) = 0.15, \quad \mathbb{P}(\gamma_{obs}) = 0.07 \tag{15}$$

These are uniformly worse than the entropy limited scenario. Using a 'law of the minimum' criterion, with $f_{\text{bio}} \propto \min(\Delta S_{\text{tot}}, \Delta C_{\text{tot}})$ interpolates between these two scenarios, based on the value of $\epsilon_C$. The best fit for this class of models is for $\epsilon_C$ to be large enough that the biosphere is entropy limited, which is precisely what we have argued that life would strive for anyway.

*2.6. Does Life Need Plate Tectonics?*

We now turn to another planetary property that may be crucial for life: plate tectonics. Though it may come as a surprise to those who have not encountered it before, plate tectonics is considered by many geologists to be essential for life on Earth for at least three reasons: first, subduction is responsible for creating the granite which comprises the continents today, and so is ultimately responsible for producing practically all land surface on Earth [34]. Secondly, even life that does not live on land ultimately is built out of materials that are eroded from the Earth's mountain ranges [35]. Thirdly, and perhaps most importantly, the silicate weathering that takes place as a result provides an additional negative feedback loop for the amount of carbon in the atmosphere, which regulates the temperature over geological timescales to a much higher precision than would have occurred otherwise [36]. In short, plate tectonics provides a "living rock" for which the stage of life is set. However, some authors, such as those of Reference [37], consider that plate tectonics may not be crucial for maintaining planetary habitability over long timescales, making this habitability criterion subject to debate.

There is every indication that plate tectonics is 'hard' to achieve, and does not seem to be the typical state for rocky planets. Firstly, none of the other rocky bodies in our solar system have plate tectonics [38]. Additionally, its presence seems to have been facilitated by a number of compounding factors: the presence of liquid water greatly increases the ductility of the crust and mantle, enabling subduction [39]. Life itself may play a critical role in speeding up the process by enhancing erosion and deposition of carbonate [40]. The fact that it relies on two different sources of internal heat, both primordial and radioactivity, of roughly equal contribution [41], could be construed as the hallmark of a selection pressure. Taken together, these strongly argue that plate tectonics is the exception rather than the rule, and is accomplished on Earth only by a plethora of independent helping factors.

All this is further compounded by the interesting coincidence: there is a narrow range of planetary masses for which plate tectonics exists, between $0.7 - 2 R_\oplus$ [42,43]. This is determined by the tuning that the convective stress of the mantle appropriately balances the lithospheric yield stress of the crust. Since mantle convection is dictated by the amount of heat contained, it is a function of the planetary mass: too small, and the crust is locked in a stagnant lid regime, too large, and it is molten. This becomes all the more intriguing when it is noted that this narrow range happens to precisely coincide with the equally narrow range of planetary masses permitting an Earthlike atmosphere, between $0.7 - 1.6 R_\oplus$ [44]. This mass range is usually taken to be important as well, under the auspices that both Marslike and Neptunelike planets are inhospitable to complex life requiring the presence of liquid water, so far as we can tell. It seems a remarkable coincidence that these two narrow windows just so happen to coincide, given the many orders of magnitude of planetary masses.

This coincidence was investigated in detail in Reference [45], where it was found that, since the radioactive heat is generated by alpha decays, which are tunneling processes, their lifetime depends exponentially on the fine structure constant (and to a lesser degree on the other parameters). If $\alpha$ were increased to a value of $1/136$, all possibly relevant alpha decays would occur with half lives of less than Gyr timescales, and so would have decayed by this point, leaving the Earth cool and

stagnant. If decreased beyond a value of 1/153, the typical timescale would be much larger, making all radioactive compounds effectively stable. Thus, if radiogenic plate tectonics is deemed important for life, the range of allowable $\alpha$ is considerably narrowed. What is more, the observed value of 1/137 is extremely close the the maximal value, indicating a strong preference for large $\alpha$. If plate tectonics is crucial, we expect a habitability criterion that reflects this, by exhibiting strong preference for large $\alpha$.

We have systematically combined the plate tectonics condition with all our previous habitability criteria to determine which combinations are consistent with the multiverse hypothesis. We report a few: if the yellow and entropy conditions are included along with the plate tectonics condition, we have the probabilities

$$\mathbb{P}(\alpha_{obs}) = 0.064, \quad \mathbb{P}(\beta_{obs}) = 0.38, \quad \mathbb{P}(\gamma_{obs}) = 0.20 \tag{16}$$

If we include the photosynthesis, entropy and tidal locking condition, we have

$$\mathbb{P}(\alpha_{obs}) = 0.063, \quad \mathbb{P}(\beta_{obs}) = 0.50, \quad \mathbb{P}(\gamma_{obs}) = 0.29 \tag{17}$$

This is not an exhaustive list: we are reaching a point where it becomes untenable to report every criterion in table format, even when restricting to those above some threshold, and so we will release the full list online as supplemental material at publication. Rather, these three representatives form germs: combinations of habitability criteria that all additional successful hypotheses will contain. If one additionally includes the convective, biological timescale, terrestrial mass, or temperate conditions to any of the above, the probabilities will be shifted slightly but the overall conclusions will still hold. This class of criteria can be said to be indifferent to these additional hypotheses.

The first thing to note is that the probability of observing our value of the fine structure constant is always diminished, since it is still rather close to the anthropic boundary. This makes the Bayesian evidence for the necessity of plate tectonics around 3–6 times weaker than for the hypothesis that it is unnecessary, which is not quite low enough to exclude this scenario.

There are a number of subtleties in the interpretation of this, however. Firstly, it is unclear whether this indicates that plate tectonics itself should be unimportant for complex life, or whether radioactivity is ultimately unimportant for plate tectonics. If the former, then we should expect to find just as much life of planets that do not support plate tectonics, be they too dry, small, large, or stiff. If the latter, then we will no doubt discover planets with perfectly active plate tectonics that are not as enriched in radioactive isotopes as ours, be that from the circumstances of their birth environment, or possibly their age.

We stress again that it will be impossible to ultimately derive a version of habitability that is uniquely compatible with the multiverse hypothesis, robust against the future inclusion of additional considerations. What we can hope to achieve is the enumeration of all possible notions that are compatible with the multiverse, and the eventual determination of which is true. Should the single true condition match any of these, it can be taken as compatible with a multiverse, and should any of the independent components of this ultimate criterion fail, it will be strong evidence against.

## 3. Why Did Life Procrastinate So Long?

In the previous section, we found habitability criteria that make our observed values of the constants typical. In order to be fully consistent, however, it was necessary to include an additional ingredient, the probability that we find ourselves in our particular location within our universe. Likewise, we may ask a similar question, the probability of finding ourselves at our particular time. Maintaining that our notion of habitability ought to account for this as well is shown to be equally constraining, and can allow us to make inferences about the distribution and frequency of life throughout our universe that we would not have been able to deduce without the added input of the multiverse hypothesis.

Namely, the conundrum we address now is the question of why we find ourselves so close to the end of our star's habitable phase, when the timescales of biological and stellar evolution are not

obviously related. Several different models have been put forward to account for this coincidence, all of which recast it as an artefact of a selection effect. The hard step model posits that the evolutionary path to intelligence required a half dozen or so incredibly difficult innovations which individually each have a very small probability of occurring within a stellar lifetime [46]. The bated breath model, on the other hand, allows that the ratio of these two timescales is a steep function of stellar mass, and so naturally most observers would arise around the smallest stars capable of giving rise to intelligent observers [47]. The easy stroll model holds that intelligence is rather reliably developed after a certain period of time, but that local planetary conditions cause the distribution of habitable lifetimes be be very steep [48].

These three models will be considered in turn. While there is no way to distinguish which is right on an observational basis at the moment, we take a different approach and ask what the compatibility of each is with the multiverse hypothesis. The first two will be shown to be incompatible with the multiverse, causing us to greatly favor the third. These will ostensibly be tested in the conventional sense eventually, allowing us to compare the prediction we make with observation.

Note that strictly speaking, the contents of this section deal more directly with the $f_{int}$ term in the Drake equation, which is the probability that a planet that has already developed life gives rise to intelligent observers. Though this factor will be the main subject of our follow-up work [49], we include this discussion here anyway.

### 3.1. Would We Live in This Universe if the Hard Step Model Is True?

The first hypothesis we consider is the hard step model, originally proposed in Reference [46]. Its tenet is that the emergence of intelligence requires a small number of very hard evolutionary innovations, each of which typically take much longer than the 5–10 Gyr timescale of stellar evolution. This scenario was studied in Reference [50], where it was found that because we are roughly 4/5 of the way through the Earth's habitable phase, the best fit value of the number of steps is 4, though within 95% confidence the possible range is between 1–16 [27]. A biological perspective was applied to try to identify what these steps could be in Reference [51] on the basis of reorganizations of information processing, and is consistent with this number, and including the distribution of these other purported hard steps in time bolsters the agreement with this model. The hard step model was combined with stellar activity models to deduce that life should be most probable around K dwarfs in Reference [52]. One important consequence of this model is that intelligent life should be quite rare in the universe, since it relies on a sequence of improbable events.

According to this model, the probability of intelligence arising on a planet after a time $t$ is

$$f_{int}(t) = \left(\frac{t}{T}\right)^{n_{hard}} \tag{18}$$

For definiteness we take $n_{hard} = 4$ throughout, and the timescale $T$ is taken to be much larger than any other that appears in the evolution of the system. In the following we may use in this expression any of the parameterizations for time which we considered: either strictly proportional to time elapsed, or else weighted by the size of the planet in question, or the size of the biosphere. This defines the pure hard step model, but a more general version will be considered after.

It is simple to see that this is incompatible with the multiverse: roughly speaking, since it greatly favors stars with the longest possible lifetimes, we would be ten thousand times more likely to inhabit a universe where stars last just ten times as long. This only exacerbates the problems we found when we considered the probability of the emergence of life to be linearly dependent on the total time. Without weighting by entropy, the probabilities we find are

$$\mathbb{P}(\alpha_{obs}) = 0.49, \quad \mathbb{P}(\beta_{obs}) = 0.009, \quad \mathbb{P}(\gamma_{obs}) = 1.1 \times 10^{-5} \tag{19}$$

Additionally, it was pointed out in Reference [27] that if the hard step model is employed, our chances of orbiting a yellow star are greatly reduced except in the case where tidally locked planets are considered uninhabitable. Accordingly, we find that $P(M_\odot) = 1.4 \times 10^{-12}$ without the tidal locking criterion, and $P(M_\odot) = 0.18$ including it. Our numbers differ from their analysis because there a more sophisticated measure of how the stellar lifetime scales with mass was used, but our simplified parameterization is sufficient to prove the point.

Since previously we had more success considering that the emergence of life should be not just dependent on the time available, but also weighted by the size of the biosphere, we may try this here, to see if it fares any better. We find that this modified version of the hypothesis $f_{\text{int}} \propto \Delta S_{\text{tot}}^{n_{\text{hard}}}$ is even more problematic, yielding

$$\mathbb{P}(\alpha_{obs}) = 0.12, \quad \mathbb{P}(\beta_{obs}) = 0.044, \quad \mathbb{P}(\gamma_{obs}) = 2.2 \times 10^{-9} \tag{20}$$

The size condition does even worse than these, giving probabilities which are indistinguishable from 0 to the 16-point numerical precision to which we work. This is so far the worst suite of hypotheses considered.

### 3.1.1. Can the Hard Step Model Work if We Are Close to the Turnover Scale?

We have seen that the hard step model as specified is drastically incompatible with the framework we are employing. The other extreme, the equihabitable condition, works much better, but this fails to explain the coincidence that it has taken approximately the full duration of the habitable time for intelligent life to arise on Earth. Before discarding this model completely, we may ask whether these two failures can be reconciled by acknowledging that they both are limiting cases of a more general probability distribution for life to arise.

Let us illustrate this in the 1 step case first, for simplicity: then the probability for life to arise on a suitable planet after a time $t$ is

$$c_1(t) = 1 - e^{-t/t_1} \tag{21}$$

As can be seen, for times much shorter than the intrinsic timescale of this distribution $t_1$, this recovers the linear dependence we saw previously. As $t$ becomes larger, the probability that life would have emerged at some point becomes more certain, eventually saturating at 1. This more general distribution then interpolates between the two cases we considered before, with the expense of adding an additional parameter.

This can be generalized to $n$ steps, by noting that the probability density function (giving the chances of a step to occur at a given moment in time) is given recursively by the formula $p_n(t) = \int_0^t dt' p_1(t - t') p_{n-1}(t')$. This yields an expression for the cumulative probability:

$$c_n(t) = 1 - \sum_{i=1}^n t_i^{n-1} Z_i e^{-t/t_i}, \quad Z_i = \frac{1}{\prod_{j \neq i}(t_i - t_j)} \tag{22}$$

In the limit that the time is much shorter than all timescales in this expression, this asymptotes to

$$c_n(t) \rightarrow \frac{t^n}{n! \prod_{i=1}^n t_i}, \tag{23}$$

which reproduces the hard step model in Equation (18) (where the factorial had been absorbed into the definition of $T$), and asymptotes to 1 in the opposite limit. It has the additional feature that it

approximates an *m* step model if *m* of the times are much greater than the timescale in question, the others much shorter. This is a consequence of the mathematical formulae

$$\sum_i Z_i t_i^k = \begin{cases} (\prod_i t_i)^{-1} & k = -1 \\ 0 & 0 \leqslant k < n-1 \\ 1 & k = n-1 \\ \sum_i t_i & k = n \end{cases} \tag{24}$$

and allows us to treat the number of critical steps as a sliding scale that depends on the time frame in question. Thus, the probability for life to emerge, for instance, on a Mars or Venus like planet, which went through a brief habitable phase in the beginning of the solar system, would not be a simple extrapolation of the 4 step model that appears to govern life on Earth, but instead would be given by a much larger number of steps. Evolutionary innovations that are trivial on the scale of millions of years become insurmountable when you only have an afternoon.

Before discussing the complication of how the timescales in this distribution are chosen, we make the simplification that they are all equal to a common timescale $T$. Then, the probability attains a highly simplified form

$$c_n(t) = \frac{\gamma(n, t/T)}{\Gamma(n)} \tag{25}$$

where $\gamma(n, x) = \int_0^x ds\, s^{n-1} e^{-s}$ is the lower incomplete gamma function. With this distribution, the expected time for the emergence of intelligent life on a planet with habitable duration $t_{\text{hab}}$, conditioned on the fact that the event does occur, is

$$t_{\text{int}} = \frac{\gamma(n+1, t_{\text{hab}}/T)}{\gamma(n, t_{\text{hab}}/T)} T \rightarrow \begin{cases} \frac{n}{n+1} t_{\text{hab}} & t_{\text{hab}} \ll T \\ n\, T & t_{\text{hab}} \gg T \end{cases} \tag{26}$$

Let us discuss the behavior of this model: its features can be roughly summarized by saying that for stars with lifetimes greater than $T$, the probability of life is a constant, whereas for those with lifetimes less than $T$ it is suppressed. We then must integrate over the distribution of stars to arrive at the probabilities for this habitability hypothesis. However, this always yields inconsistent results, which interpolate between the pure hard step model of Equation (20) and the photosynthesis criterion, both of which are in conflict with the multiverse. So perhaps unsurprisingly, given the results of the previous section (which would correspond to the $n_{\text{hard}} = 1$ model), taking the entropy produced by the sun as close to the threshold to guarantee that life arises does nothing to rescue the hard step model's incompatibility with the multiverse hypothesis.

### 3.1.2. Disparate Timescales

Previously, we made the simplification that all critical step timescales were the same, in order to simplify the expressions needed. This is certainly an unwarranted approximation; here we rectify this, and show that there is even more reason to disfavor this model.

What is needed is the underlying distribution of timescales for biological innovations to take place. This can be determined by extrapolation: since life on Earth has developed a whole suite of innovations throughout its history, statistics can be performed on the relatively more mundane ones that took place, and used to determine the underlying probability for an innovation to take a given amount of time. We use the list compiled in Reference [53], where the origination of 60ish innovations of higher organisms are tabulated. Taking rank-order statistics of the time difference between successive innovations, as displayed in Figure 5, yields a cumulative distribution function consistent with a power law of slope 1/4,

$$c(t) = \left(\frac{t_{\text{cut}}}{t}\right)^{1/4} \tag{27}$$

Restricting to our lineage instead leads to a slope of 1/2, but the difference between these two is inconsequential, so we specify to the former. This also requires a cutoff timescale $t_{cut}$, which specifies what is to be considered a 'hard' innovation. In the following, we take $t_{cut} = 50$ Myr, but the results are rather insensitive to the exact number used. With these choices, the number of hard steps will be given by a binomial distribution

$$p(N_{hard}) = \text{Binomial}\left(\frac{4}{c(t_{\oplus})}, c(t_{\oplus})\right) \tag{28}$$

So that the expected number of hard steps will be $\langle N_{hard}\rangle = 4 \pm 2\sqrt{1 - c(t_{\oplus})}$, the variance being $\sigma_{N_{hard}} = 1.65$ for our choices. We have normalized the mean to be the most likely value for our Earth system, for definiteness.

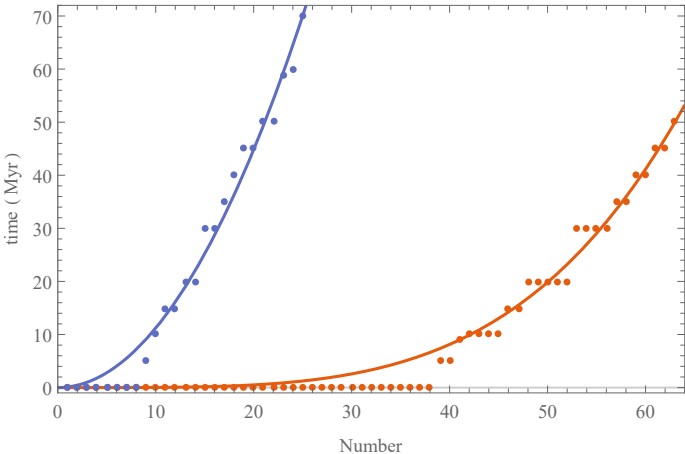

**Figure 5.** The observed distribution of innovation timescales, along with the best fit curves. Here, blue specifies to innovations occurring in our lineage, and orange to innovations across all lineages.

This raises an important criticism of the hard step model: if the timescales are all independent identically distributed variables drawn from this random distribution, then it is fairly likely for a random instantiation to have relatively fewer hard steps. This must then be coupled to the steep suppression of the emergence of life on systems with a larger number of critical steps. In this view, we would be overwhelmingly likely to have developed on a planet with an unusually small number of critical steps, rather than arising as one of the extraordinarily lucky representatives with an average number of steps. This argument is independent of multiverse reasoning: as long as there is any distribution for the hardness of evolutionary innovations, it is much easier for a planet to be accidentally easier at facilitating the development of intelligent species than it is for intelligence to arise on an average planet. Based off this consideration alone, we should not expect the hard step model to be a good description of the distribution of intelligent life within our universe. Of course, some innovations, such as the evolution of oxygenic photosynthesis and eukaryogenesis, may belong to a qualitatively more difficult class, in which case the hard step model preserves its explanatory character.

### 3.2. Was Complex Life Waiting for Earth to Oxygenate?

A second popular explanation for why complex life only started around 600 Mya is that it was necessary for the Earth's atmosphere to fully oxygenate first [54]. This increased the energy available for metabolic processes, which subsequently became indispensable for animal life, and also created an ozone shield that made the colonization of land possible. In this framework, the reason complex life took billions of years to get started is simply due to the fact that that is how long it took the Earth to oxygenate. Upon inspection, though, this explanation raises the secondary conundrum as to why it takes almost precisely the stellar lifetime for this to occur; this has not been answered conclusively.

Here we detail several leading mechanisms for what triggered this, and how they fit in with anthropic selection pressures.

Though photosynthetic life is usually implicated in the oxygenation of Earth, this alone would not contribute a significant amount of oxygen if counterbalancing oxygen sinks adjusted to absorb it [55]. Additionally, the innovation of oxygenic photosynthesis cannot be the complete mechanism, as there is evidence [56] that this evolved at least 500 Myr before the first increase in oxygen levels 2.2 Gya. Even after the original oxygenation event, levels stalled at perhaps 1% of current levels until 600 Mya, when oxygen levels reached roughly their present values [57], so the long delay must have been due to significant oxygen depletion that kept pace with the biological production rates [58]. This depletion is to be expected, as at the time both the Earth's crust and atmosphere would have been strongly reducing, soaking up any oxygen that was produced. It was only after enough had been injected into the system that these oxygen sinks would have depleted, finally allowing for a buildup of the gas to form. These sinks can be classified into two broad types: either the reductants would have been depleted by being drawn down, into the Earth's mantle, or up, into space. The fact that the relative rates of each of these processes is uncertain to within an order of magnitude [59] leads to uncertainty over which was dominant. We consider each in turn.

### 3.2.1. Drawdown

The most standard explanation for the oxygenation time is that the Archean Earth was reducing, with plenty of raw iron and sulfur in the crust that would have immediately neutralized any excess oxygen that appeared in the atmosphere [60]. It was only once this initial reservoir was depleted that oxygen could build up to its current state. In order to estimate the time it would take for this transition to take place, we can compare the rate of drawdown to the total mass of the atmosphere to find

$$t_{O_2\downarrow} \sim \frac{M_{\text{atm}}}{\Gamma_{\text{down}}} \tag{29}$$

If we know how both of these quantities scale with parameters, we can use this to determine whether this timescale is naturally of the order of the stellar lifetime. Of the two, the drawdown rate is on relatively firmer footing; this is because there is no clear explanation for the mass of the Earth's atmosphere.

There are various schools of thought as to why the mass of our atmosphere should be $M_{\text{atm}} \sim 10^{-6} M_\oplus$. Even within our own solar system, there is a very large spread in the ratio of atmospheric mass to planet mass, so atmosphere is likely to be highly variable and sensitive to local conditions [61]. Given that the source of Earth's atmosphere seems to have important contributions both from planetary outgassing and delivery from asteroids or comets [62], the distribution is highly uncertain. This uncertainty is compounded by the fact that we still do not know what the range of habitable atmospheric masses is. Here, we will follow the expectation that there is a narrow window of habitable atmospheric masses, based off the pressure at the surface.

The total mass of the atmosphere can be related to the pressure at the surface by the expression $M_{\text{atm}} = 4\pi R_{\text{terr}}^2 P_{\text{atm}}/g$. Noting that the surface pressure of Earth is two orders of magnitude larger than the minimum required for the presence of stable liquid water, $P_{\text{trip}} = 0.006$ atm, we use this as a guiding value to set the atmospheric mass necessary in alternate universes. In fact, this could easily be a natural state of affairs: the closer the atmosphere is to the triple point, the more susceptible it would be to climate fluctuations that could lead to a runaway icehouse or greenhouse scenario. We do not pursue this line of reasoning quantitatively here, but use it to argue that it is plausible that the smallest possible atmosphere would be an order of magnitude or two above the triple point of water.

It still needs to be explained why we would be situated near the minimal value, if larger atmospheres are favored on climate stability grounds. One possibility would be that the atmospheric mass distribution could be very steep, greatly favoring smaller values. Alternatively, it may be that smaller atmospheres are more conducive to life. Earth's usual biochemistry ceases to operate in regions of extreme pressure. This is especially true of the lipid chemistry that is essential for the functioning

of cell membranes- at high pressures, the membrane stops behaving as a two dimensional fluid surface [63]. On the other hand, the piezosphere, the regions of the Earth with unusually high pressure such as deep in the ocean and within the crust, is not completely devoid of life. There, piezophiles have adapted to their environment by using unsaturated fatty acids, which are more 'knobby' that the simple rod shaped saturated ones normally employed, as to resist jamming [64]. These types of adaptations even allow for complex animal life deep within the Mariana trench. There is certainly less biological activity in these realms, but it is challenging to attribute this to the extreme pressures, as these regions also have reduced nutrient flux, lower light, and lower temperature [65]. In light of this, it is not clear that a significantly larger atmosphere would pose much of an evolutionary challenge to life. However, perhaps it would lead to a significant lengthening of evolutionary timescales, and this is the explanation for the atmospheric mass we observe. In this case, the range of habitable atmospheric masses would be quite sharp, and so we would be justified in fixing $P_{\text{atm}} \sim 100 P_{\text{trip}}$. We follow this route in this paper, though it certainly would be interesting to explore these ideas about atmospheric mass more fully, and the implications they have for the distribution of both atmospheres and life throughout the universe.

This then raises the question of how the triple point of water depends on fundamental constants. The solid-liquid transition is practically independent of pressure, and occurs when the temperature is high enough to excite vibrational modes of the water molecules. This energy is given by $E_{\text{vib}} \sim T_{\text{mol}} = 0.037\alpha^2 m_e^{3/2}/m_p^{1/2}$. The water-gas phase curve is given by the Clausius-Clapeyron equation as $P(T) = P_0 e^{-L/T}$. The latent heat of evaporation is equal to the intermolecular binding energy, $L \sim \alpha/r_{\text{H}_2\text{O}}$. The coefficient $P_0$ is an integration constant that in principle can be derived from a statistical mechanical treatment capable of yielding an exact expression for the chemical potential of liquid water, but the author is not aware of progress in this direction, so the phenomenological value $P_0 = 1.3 \times 10^5 T_{\text{mol}}/r_{\text{H}_2\text{O}}^3$ will be used instead. This aesthetic choice will not strongly affect the calculation, since the exponential dependence plays the dominant role in the scaling with parameters. Using $r_{\text{H}_2\text{O}} = 5.9 a_0$, where $a_0$ is the Bohr radius, we find

$$P_{\text{atm}} = 22.8 \, \frac{\alpha^5 \, m_e^{9/2}}{m_p^{1/2}} \, e^{-0.44 \sqrt{\frac{mp}{me}}} \tag{30}$$

From here, the rate of drawdown of $O_2$ from the oxidation of eroded material must be calculated to determine the timescale. We use Equation (A10) from the Appendix, which for our current atmospheric mass gives several Gyr as an oxidation time.

In relation to the lifetime of the host star, this yields

$$\frac{t_{O_2\downarrow}}{t_\star} = 341 \, \lambda^{5/2} \, \alpha^{-3} \, \beta^{1/4} \, e^{-0.44\beta^{-1/2}} \tag{31}$$

This is normalized to be 1 Gyr for sunlike stars for definiteness, giving a ratio of about 0.2, in agreement with Reference [66]. Demanding that this quantity be less than 1 for a planet to be habitable gives an upper bound on the mass of the star, which is in contrast to the requirement that arises if the oxidation were due to escape to space.

The aim of this model was to explain the ratio of the two timescales, despite their drastically different physical origins. Even though this scenario favors large mass stars, the scaling is very close to the Salpeter slope in the initial mass function, so that in the equihabitable scenario the observed value is not so unlikely. Even in the entropy-weighted scenario, this ratio can usually be close to 1, provided that there is a cutoff in habitable stellar masses close to the solar value. We quantify this in two ways: the first is $P(r_t > 0.2)$ restricting to our observed values of the physical constants, and the second is $\mu_{r_t} \pm \sigma_{r_t}$, again within our constants. In Table 2, we find that with either statistic these ratios are quite compatible with observation. Of note is that the first is essentially equivalent to the probability of

orbiting a star of our sun's mass, as long as no high mass cutoff is introduced, as in the biological timescale criterion.

**Table 2.** Statistics for the ratio of oxygenation time to the stellar lifetime, both within our universe (subscript *U*) and in the multiverse (subscript *M*). All figures are computed with the entropy+yellow criteria. Even though they all satisfactorily explain our observed ratio of about 0.2 when restricted to our universe, none of them explain why we do not live in another universe where the average ratio is much smaller. Note that these values are optimistic: if the observed ratio is taken as being any higher, the probabilities will be even lower.

| Mechanism | $P(r_t > 0.2)_U$ | $(\mu_{r_t} \pm \sigma_{r_t})_U$ | $P(r_t > 0.2)_M$ | $(\mu_{r_t} \pm \sigma_{r_t})_M$ |
|---|---|---|---|---|
| drawdown | 0.38 | $0.22 \pm 0.07$ | 0.03 | $0.03 \pm 0.08$ |
| drawup | 0.49 | $0.24 \pm 0.14$ | 0.05 | $0.04 \pm 0.11$ |
| combined | 0.56 | $0.20 \pm 0.04$ | 0.05 | $0.004 \pm 0.10$ |

This appears to be a valid explanation of why life took so long to develop. However, if we employ the multiverse hypothesis, we can add additional statistics by letting the physical constants vary. There, we find that this ratio is actually much smaller than one in a sizable fraction of universes. The majority of universes, in fact, we find will oxygenate their planets much more quickly than ours, producing a paradox of why we would have ended up in this one.

In contrast, the oxygenation delay does not appreciably alter the probabilities of being in our universe, as displayed in Table 3. So, in contrast to the other hypotheses considered in this paper, this one is incompatible with the multiverse on the basis that its purported explanatory powers are undermined when the two ideas are combined. This will hold for the alternative oxygenation mechanism as well, as we now discuss.

**Table 3.** The probabilities with various oxygenation mechanisms. All utilize the entropy + yellow criteria.

| Mechanism | $\mathbb{P}(\alpha_{obs})$ | $\mathbb{P}(\beta_{obs})$ | $\mathbb{P}(\gamma_{obs})$ | $P(M_\odot)$ |
|---|---|---|---|---|
| drawdown | 0.187 | 0.487 | 0.317 | 0.384 |
| drawup | 0.279 | 0.172 | 0.430 | 0.507 |
| combined | 0.190 | 0.494 | 0.322 | 0.443 |

### 3.2.2. Drawup

An explanation for the coincidence of these two timescales was proposed in Reference [47], on the basis that their ratio $t_{\text{life}}/t_\star$ may decrease with stellar mass. Then, since the stellar mass distribution is so steep, we would naturally expect to be situated around a star just large enough to barely satisfy the requirement $t_{\text{life}} \sim t_\star$. If the oxygenation of the Earth were dependent on stellar activity this would naturally fit with this explanation, as this process would then take longer around smaller stars.

This mechanism was proposed in Reference [59], where atmospheric reductants are eventually lost to space. Here, geochemical processes would have dissociated methane and water molecules, followed by the escape of the hydrogen. This leads to an imbalance of carbon, which combines into carbon dioxide that is then drawn down into the mantle. The escape process is limited by the UV flux of the star, which depends on stellar mass, and only matches the stellar timescale around sunlike stars.

We are not in a position to judge this hypothesis based off its geological merit, but we may consider the implications it has on the distribution of observers throughout the multiverse. If this process is the limiting factor for where life can arise, then we would expect to be unable to find universes that can oxygenate planets much faster than our own. In the following we specify to main sequence stars, disregarding any enhanced atmospheric erosion that would occur during flares of young stars. This has recently been a topic of intense interest [67], and may play an important role in determining the habitability of planets around red dwarfs [68].

The UV process is dominated by photons that can just barely cause hydrogen to escape the atmosphere, since those with higher energies are exponentially suppressed. Then, to estimate the time required to completely oxygenate a planetary atmosphere, the amount of flux in this range must be inferred. For a blackbody at temperature $T$, this would be

$$f_{\mathrm{XUV}} \approx \frac{1}{2} \left( \frac{E_{\mathrm{XUV}}}{T} \right)^2 e^{-E_{\mathrm{XUV}}/T}, \tag{32}$$

where $E_{\mathrm{XUV}} \sim 10$ eV. Stars are not blackbodies in this energy range, but the flux can be estimated as being a factor of 20 higher than the black body flux [69]. For the early Earth, the sun's flux implied a photolysis rate of $10^{12-13}$ mol/year [59]. (It is worth noting that the oxidation rate is much smaller currently, due to the presence of cold traps and other concentration effects that prevent dissociable compounds from reaching the Earth's exobase. Additionally, planetary characteristics such as a magnetic field can prevent atmospheric loss, leading to uncertainties in the overall rate [70,71].) To first approximation, about one hydrogen ion escapes for every UV photon incident, and so the oxygenation timescale is then given by

$$t_{\mathrm{O_2}\uparrow} = \frac{M_{\mathrm{atm}}/m_p}{\Phi_{\mathrm{XUV}} R_{\mathrm{terr}}^2} \tag{33}$$

Here $M_{\mathrm{atm}}$ is the mass of the atmosphere and the flux is given by $\Phi_{\mathrm{XUV}} = 20\, T^3\, f_{\mathrm{XUV}}$, so that the ratio of timescales is

$$\frac{t_{\mathrm{O_2}\uparrow}}{t_\star} = 1.1 \times 10^9 \frac{\beta^{3/4} \gamma^{3/4}}{\alpha^4} \lambda^2 \hat{e} \left( -\frac{0.44}{\sqrt{\beta}} + 841 \frac{\alpha^{3/2} \beta}{\gamma^{1/4} \lambda^{1/2}} \right) \tag{34}$$

Though the exponential dependence on mass differs from the usual power law form found in the literature, around solar mass values this function behaves with effective power law index $p \equiv d\log(t_{\mathrm{O_2}}/t_\star)/d\log(\lambda)$, which ranges from $-8.1$ to $-3.2$ within a factor of two of solar mass. This can be compared to the estimates of $p = -3.4$ from Reference [72] and $p = -6.6$ of Reference [47], and a seeming $-4.25$ from Reference [73].

For life to develop, this ratio must necessarily be less than 1. This will be the case for intermediate values of $\lambda$, constants permitting. For small masses, the stellar temperature is so low that photons capable of ejecting hydrogen from the atmosphere are infrequent, leaving the gas trapped and unable to oxidize. For large masses, the stellar lifetime is very short, leading the star to burn out well before enough hydrogen has escaped. To first approximation, these delineating masses are given by

$$\lambda_{\min} \approx 3.7 \times 10^6 \frac{\alpha^3 \beta^3}{\gamma^{1/2}}, \quad \lambda_{\max} \approx 3.0 \times 10^{-5} \frac{\alpha^2}{\beta^{3/8} \gamma^{3/8}} e^{22/\sqrt{\beta}} \tag{35}$$

The former corresponds to $0.66 M_\odot$ in our universe, and the latter is irrelevantly large. (The exact expressions can be given in terms of Lambert productlogs, but the error from these approximations are only a few percent.) The minimum of the ratio of timescales will be larger than 1 when

$$\frac{\alpha^2 \beta^{19/4}}{\gamma^{1/4}} e^{-0.44/\sqrt{\beta}} < 8.5 \times 10^{-21}. \tag{36}$$

Finally, let us comment on the possibility that both drawup and drawdown are relevant for setting the oxygenation timescale. In this case, we would have

$$\frac{1}{t_{\mathrm{O_2}}} = \frac{\epsilon}{t_{\mathrm{O_2}\downarrow}} + \frac{1-\epsilon}{t_{\mathrm{O_2}\uparrow}} \tag{37}$$

where the two timescales are given by Equations (31) and (34) above, and $\epsilon$ is a free parameter between 0 and 1 that dictates the relative importance of each process. Taking both contributions to be equal is problematic, for as we have seen the rate of drawdown is higher for low mass stars, and the rate of

drawup is higher for large mass stars. In this case, then, every star's oxygenation time is smaller than its lifetime, with a switchover in the dominant mode occurring for intermediate masses, as shown in Figure 6. Generically, the presence of drawdown serves to spoil the explanation for the observed coincidence of timescales, as it removes the low mass cutoff. The only way for this not to occur is for sufficiently small values of $\epsilon$: for $t_{O_2\downarrow} \gtrsim 18 t_\star$, a range of stellar masses has a ratio which is larger than 1. However, when considered from a multiverse perspective, most observers would still not expect these two timescales to coincide on their planet. Therefore, the multiverse gives reason to disfavor planetary oxygenation as the mechanism for the delay of complex life. This is an otherwise perfectly viable hypothesis, and if it does turn out to be true, we will have strong evidence against the multiverse.

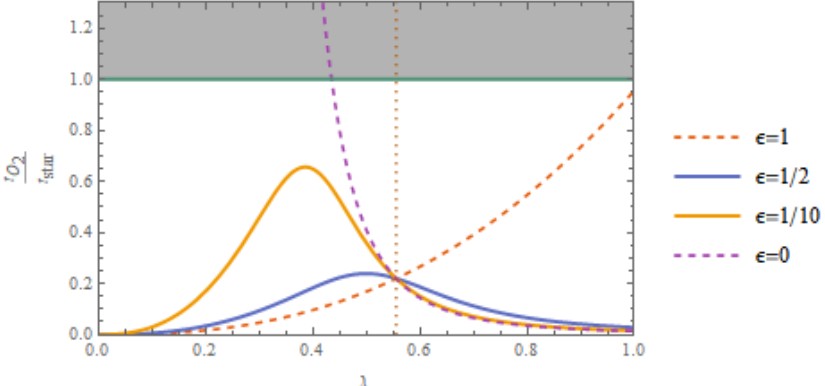

**Figure 6.** Ratio of oxygenation time to stellar lifetime as a function of stellar mass. For this criterion, stars with ratio above 1 are uninhabitable. Here, $\epsilon$ parameterizes the relative importance of drawdown and drawup. The dotted line corresponds to 1 $M_\odot$.

*3.3. Easy Stroll*

The remaining account for the coincidence of solar and biological timescales is known as the easy stroll model [48]. In this model the distribution of planetary habitable durations is relatively decoupled from the lifetime of the host star, with a steep preference for very short duration. If we take the planetary histories of Venus and Mars into account, for example, it becomes easy to imagine that the majority of planets, born even within the temperate zone, will through runaway processes lose their clement nature on the order of a geological timescale [74]. It has been proposed that the presence of life itself can temper some of the most dangerous negative feedbacks, leading to a bottleneck amongst worlds where this large scale alteration initially takes hold [75].

In this model, developing complex life is relatively easy, being perhaps proportional to the total lifetime (weighted appropriately) as in Section 2, but the planetary maintenance of a habitable phase becomes the significant bottleneck. Since the majority of planets will have habitable lifetimes much too short for complex life to develop, we would then naturally expect to arise on a planet that is quite close to its expiration date.

This state of affairs effectively adds a hidden variable that we do not consider in our simplistic account, which dwells only on stellar mass. This may be planetary mass, composition, volatile abundance, orbital parameters, or any number of other things. The prediction of this model is that the distribution for at least one of these will be important for habitability, and also sufficiently steep so that the ratio of timescales is typically on the order of the threshold value. Exploring this prediction in detail will need to remain for future work.

## 4. Discussion: Comparing 10,560 Hypotheses

In the multiverse setting, we expect to live in a universe that is good at producing observers. There are undoubtedly many conditions that must be met for a universe to be able to achieve this feat, but at the moment we do not know what they are. As such, there are a vast number of hypotheses in

the literature, some complementary, others mutually contradictory. By tabulating all of these, it will be possible to delineate which are compatible with the multiverse expectation that our universe is an exceptional observer factory, and which are not. As of now, each habitability criterion is strictly hypothetical, but one day we will definitively know what life needs. Once this is established, it can be used as either negative evidence against the multiverse, or positive evidence for the multiverse. This rather baroque procedure is necessary to circumvent the major charges against this scientific paradigm, that it does not lead to any directly observable consequences. This reliance on indirect means in no way diminishes this framework, as in fact the act of doing so allows many predictive and explanatory statements to be made.

Of course, if I tell you that the multiverse dictates that complex life should need photosynthesis, and we later verify that this is in fact true, it would be a tremendous overstatement to say that the multiverse is anywhere close to being proven right. The opposite situation is much less forgiving, mind you: if the prediction is wrong, we can safely forget the idea of other universes, and focus on explaining why ours is unique. However, the key to overcoming this rather weak positive evidence, which on the face of things boils down to something like a 50/50 chance of being right, is the fact that statements can be made about a very large variety of potentially relevant conditions for life. Taken all together, this can amount to perhaps a dozen independent predictions for what life needs. By marshalling these various predictions, the even split from a lone condition turns into a 1 in a 1000 chance of getting all of them right. This could be compared to the scenario where the multiverse is a fiction, in which case we would expect roughly half the predictions to be false.

The challenge here is that the different predictions are not exactly all independent: as they add various anthropic boundaries and preferences for specific parameter values, there can be a nontrivial interplay between the different habitability conditions. We have seen some of this already, as for example when the plate tectonics condition lead to acceptable probabilities for only a subset of the hypotheses. Because of this, it becomes necessary to check each combination individually, to ensure that no acceptable combinations are being overlooked. The number of possible combinations quickly proliferates, however: even with the dozen or so conditions we have considered in the three papers on the subject, there are over 10,000 combinations. What is more, there are many more relevant habitability conditions we have not even attempted to incorporate yet. Though incorporating each new criterion has already been made as streamlined as deemed possible through the python code available at https://github.com/mccsandora/Multiverse-Habitability-Handler, it still takes a laptop about a minute to check each condition to a reasonable accuracy. Even with a drastic increase in computing power, it will soon become infeasible to check every individual hypothesis.

It is important to extract general, qualitative features about how each notion of habitability affects the distribution of observers throughout the multiverse. Armed with these, it becomes possible to build an intuition as to how they will combine to deliver the ultimate figures of merit in our framework, those probabilities of observing our values of the constants.

Though we have focused on the fraction of planets which develop life in this work, we are very far from having done this topic justice. Our estimates of which planets can give rise to life has been very rudimentary and broad-brush, neglecting very many features that are probably extremely important. However, we hesitate to be too apologetic for this omission: what has been provided is a framework that can be readily extended to include arbitrary habitability criteria. The reasons for including more are twofold: first, the greater the input, the greater the output. Each habitability condition represents a potential check of the multiverse framework, and though not all will give rise to concrete predictions, incorporating as many as possible will lead to the strongest possible attempt at testing this idea. Secondly, it will be necessary to ensure that nothing is overlooked. If some habitability condition is passed over because it greatly favors some region of the multiverse which later turns out to be sterile for completely unrelated reasons, we will be in danger of drawing false conclusions. This is the challenge that presents itself, and if we want to be able to utilize this reasoning effectively it will need to be dealt with to the best of our ability.

We regard the results we have found so far as encouraging. Out of the various potentially reasonable habitability hypotheses we have discussed, the linear dependence on entropy was a clear winner. It is worth reiterating that in our assessment we used not only the probabilities of observing the values of the constants we see, but also several other local conditions, namely the mass of our star and our current moment most of the way through the Earth's habitable phase. Utilizing these additional pieces of information highlights a complementarity in our approach: some criteria that have no trouble explaining our position within our universe are dramatically incompatible with the multiverse, and others that are compatible with the multiverse cannot be reconciled with our position within our universe. It is necessary, within a multiverse framework, to be able to explain both simultaneously, and this more rigorous standard is capable of more effectively pruning the potential habitability criteria than either alone. Though we have focused our attention on a relatively few number of parameters, this procedure can eventually be done for the entire suite of both physical constants and environmental variables, which will ultimately be necessary to fully test this framework. This is going to be a tremendous challenge that will require synthesizing knowledge from a great variety of fields, but the promise of being able to fully determine whether there are other universes out there beyond our horizon will make this herculean task well worth the effort.

**Funding:** This research received no external funding.

**Acknowledgments:** I would like to thank Ileana Pérez-Rodríguez for useful discussions.

**Conflicts of Interest:** The author declares no conflict of interest.

## Appendix A. Some Geology

Many calculations throughout the text rely on the characteristic size and timescales of terrestrial planets. Here we estimate these, and determine how they depend on the physical constants.

Firstly, the mass and radius of a terrestrial planet are given by the condition that the escape velocity must be slightly greater than the thermal velocity [22]:

$$M_{\text{terr}} = 91.9 \, \frac{\alpha^{3/2} \, m_e^{3/4} \, M_{pl}^3}{m_p^{11/4}}, \quad R_{\text{terr}} = 3.6 \, \frac{M_{pl}}{\alpha^{1/2} \, m_e^{3/4} \, m_p^{5/4}} \tag{A1}$$

These have been normalized to Earth's values and we have used that the density of matter is $\rho \sim \alpha^3 m_e^3 m_p$.

The typical mountain height can be estimated by equating the molecular and gravitational energies, $H_{\text{mountain}} \sim E_{\text{mol}}/(g m_p)$ [76], to give

$$H_{\text{mountain}} = 0.0056 \, \frac{M_{pl}}{\alpha^{1/2} \, m_e^{3/4} \, m_p^{5/4}} \tag{A2}$$

This scales in the same way as the terrestrial planet radius, which is also set by balancing gravitational and molecular energy, but is 600 times smaller (10 km).

Many properties on Earth, such as the speed of continental drift and erosion rates, are determined by the planet's internal heat. A rough estimate of this can be given by dimensional analysis as

$$Q \sim G \, M_{\text{terr}} \, \rho \, \kappa_{\text{heat}} = 92.5 \, \frac{\alpha^{9/2} \, m_e^{7/2} \, M_{pl}}{m_p^{5/2}} \tag{A3}$$

Which is normalized to the observed value of 47 TW. Here, we made use of the thermal diffusivity of rock $\kappa_{\text{heat}} \sim c_s L$, where $L$ is the size of a cell in the solid, which in rock just scales with the Bohr radius. and $c_s \sim \sqrt{E_{\text{vib}}/m_p} \sim 6$ km/s is the speed of sound, yielding $\kappa_{\text{heat}} = 2/(m_e^{1/4} m_p^{3/4})$.

A more sophisticated estimate of the Earth's heat, including its time dependence, will be necessary for some applications. This problem is somewhat muddied because there are two relevant sources: the primordial heat of formation, and that released by radioactive decay. The relative importance of each was estimated from other bodies in the solar system in Reference [77], and a measured from geoneutrino flux in Reference [41]. Remarkably, these indicate that the Earth's heat budget is split almost equally between radioactivity and primordial heat. This itself is a startling coincidence, but its only relevant consequence for our present purposes is that we must calculate both contributions to the heat.

First, the primordial heat: the internal temperature is set by the gravitational energy of the planet's formation, which is given by

$$T_i \sim \frac{G\, M_{\text{terr}}\, m_p}{R_{\text{terr}}} \sim 7600 \text{ K} \tag{A4}$$

Crucially, this is above the melting temperature of rock, which led to an initially molten Earth, and its subsequent differentiation into mantle and core. This is generic for terrestrial planets: since the gravitational energy is a bit lower than the molecular bond energy to retain gases and liquids, we always have $T_i \propto T_{\text{mol}}$ for any values of the fundamental constants. Since $T_i$ is an order of magnitude larger, terrestrial planets will always be predisposed to differentiation. This carries many potential benefits, such as a magnetic field and the sequestration of highly reducing iron minerals.

The mantle's heat conductivity is much higher than garden variety rocks, on account of the dominant method of heat transfer being by convection rather than conduction. It is the conductivity of the lithosphere, the Earth's rigid outer skin, which serves as the last line of defense against the emanation of heat to space, and so it will be crucial to determine what sets the lithosphere's thickness and conductivity. From Reference [78], If the Earth is modeled as an inner mantle with infinite conductivity, and the upper lithosphere as having having diffusivity $\kappa_{\text{heat}}$, then the total heat radiating through the surface at time $t$ will be

$$Q_{\text{form}} = 4\pi\, R_{\text{terr}}^2\, \frac{\kappa_{\text{heat}}\, T_i}{L_{\text{lith}}(t)}\, \hat{e}\left( \frac{-3\, \kappa_{\text{heat}}\, t}{R_{\text{terr}}\, L_{\text{lith}}(t)} \right) \tag{A5}$$

The depth of the lithosphere increases with time as $L_{\text{lith}}(t) = 2\sqrt{\kappa_{\text{heat}} t}$. We then arrive at

$$L_{\text{lith}} = 1.56\, \frac{t^{1/2}}{m_e^{1/8}\, m_p^{3/8}} \tag{A6}$$

In terms of constants, we find

$$Q_{\text{form}} = 237\, \frac{\alpha^{9/2}\, m_e^{7/2}\, M_{pl}}{m_p^{5/2}}\, \frac{1}{s}\, e^{-s}, \quad s = 10\, \frac{\alpha^{1/2}\, m_e^{5/8}\, m_p^{7/8}\, t^{1/2}}{M_{pl}} = \left( \frac{t}{2.5 \times 10^9 \text{ yr}} \right)^{1/2} \tag{A7}$$

If we neglect the $s$ dependence of this expression, we find the scaling from above based off simple dimensional analysis.

Now, the radiogenic component: for a planet with multiple radioactive species, the heat generated by decay is given by a sum of their individual contributions. The total is then

$$Q_{\text{rad}} = \sum_i \frac{f_i\, \epsilon_i\, M_{\text{terr}}}{\tau_i}\, e^{-t/\tau_i} \tag{A8}$$

where $f_i$ is the fraction and $\epsilon_i$ the binding energy (in units of $m_p$) of species $i$.

The rate of continental drift can be found in Reference [79]. If we use the rough estimate for the planet's heat, we have

$$v_{\text{drift}} = \frac{Q}{4\pi\,R_{\text{terr}}^2\,n\,(L + c_p\,\Delta T)} \sim 1376\,\alpha^{1/2}\,\beta^{1/2}\,\gamma \sim \frac{\text{cm}}{\text{yr}} \tag{A9}$$

Here $n$ is the number density of mantle, $L$ is the latent heat, and $\Delta T$ is the difference between melting and surface temperatures, which can both be estimated as $L \sim \Delta T \sim T_{\text{mol}}$.

The weathering rate can be estimated as

$$\Gamma_{\text{drawdown}} \sim v_{\text{drift}}\,H_{\text{mountain}}\,R_{\text{terr}}\,n = 8.9 \left( \frac{\alpha\,m_e}{m_p} \right)^{5/2} M_{pl} = 5.4 \times 10^{13}\,\frac{\text{mol}}{\text{yr}} \tag{A10}$$

which agrees with observed rates.

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
