# Peer review of "Multiverse Predictions for Habitability: Fraction of Planets that Develop Life"

_universe, doi:10.3390/universe5070171_

Reviewer 1 Report

This is an interesting and comprehensive work that examines how the multiverse hypothesis may be falsified through the study of biosignatures in the future. I believe that the manuscript merits publication in the journal, but I would like to see a few points explained better.

(a) On pg. 4, the ansatz considered is proportional to the stellar lifetime. However, over a given fraction of the stellar lifetime, it is possible that habitability has a non-linear dependence on the age. For example, the species richness (no. of species) might be very roughly modelled as an exponential up to some cutoff time as suggested in Lingam & Loeb (2017), which can be cited here. Perhaps the author can comment on what will happen in case an exponential dependence on the total stellar lifetime (or a constant fraction of it) is introduced.

Lingam, M. and Loeb, A. (2017). Reduced Diversity of Life around Proxima Centauri and TRAPPIST-1. Astrophys. J. Lett., 846, L21

(b) On pg. 5, the author assumes that the fraction of stars with planets of similar size (and presumably temperature) is not dependent on stellar mass. This is a fairly good assumption, and I'd recommend citing Table 1 of Kaltenegger (2017) here.

Kaltenegger, L. (2017). How to characterize habitable worlds and signs of life. Annu Rev. Astron. Astrophys., 55:433

(c) On pg. 6, just below Eq. (4), a simple physical calculation along the lines of Weisskopf's analysis demonstrates that H_clay is inversely proportional to g. However, the density of all terrestrial planets is not quite the same, owing to which g does not scale linearly with R. Instead, it scales roughly as R^(1.7) because M is proportional to R^(3.7) as shown in the simulations by Zeng et al. (2016). Hence, this correction should be noted and the text altered accordingly.

Zeng, L., Sasselov, D. D., and Jacobsen, S. B. (2016). Mass–radius relation for rocky planets based on PREM. Astrophys. J., 819, 127

(d) Eq. (8) is accurate because the temperature of the planet is much smaller than that of its host star. Thermodynamic efficiency a la Carnot is also close to 100% and enables the effective utilization of photons. However, when it comes to planets around cool brown dwarfs, one cannot automatically assume that T_planet is much lower than T_BD. Of course, the author need not mention this aspect since they are dealing with stars.

(e) In principle, there do not seem to be strong a priori reasons, from the perspective of evolving photosystems (with reaction centres and light-harvesting complexes), that prevent the use of wavelengths<400 nm="" and="">1100 nm, especially the latter. Multi-photon capture schemes are theoretically feasible, as outlined in Wolstencroft & Raven (2002) and subsequent papers. It may be worth mentioning this cautionary statement on pg. 8.

Wolstencroft, R. D. and Raven, J. A. (2002). Photosynthesis: Likelihood of Occurrence and Possibility of Detection on Earth-like Planets. Icarus, 157:535

(f) In pgs. 11 and 12, the importance of the weathering rate is rightly pointed out as it provides the supply of nutrients. However, the steady-state concentration depends not only on the influx of nutrients but also on their characteristic residence times in the environments under question. For example, in the oceans, this is constrained by the hydrothermal circulation time, which in turn depends on both the degree of internal heat as well as a number of geochemical factors. As it is virtually impossible to model the underlying complexities of nutrient concentrations, I suggest that the author should briefly note that nutrient turnover timescales and their accompanying phenomena are not taken into account herein.

(g) With regards to plate tectonics on pg. 14, the most recent numerical models indicate that stagnant-lid tectonics can support "habitable" conditions for Gyr timescales, and that plate tectonics may not be necessary for the stabilization of climate. See, for example, Foley (2019) and references therein:

Foley, B. J. (2019). Habitability of Earth-like stagnant lid planets: Climate evolution and recovery from snowball states. arXiv:1903.12111

(h) The discussion at the beginning of pg. 18 was interesting, and I'd appreciate it if the author could clear up a certain point for me. Suppose, for the sake of argument, that one subscribes to a "strong" version of evolutionary convergence wherein the PDF for the number of hard steps is close to a delta function centered on the no. of hard steps required on Earth. What would this imply for the likelihood of intelligence to emerge on an average planet?

(i) On pg. 21, subsequent numerical simulations after Airapetian et al. (2017) [Ref. 62] have shown that the retention of atmospheres on Gyr timescales might be possible in some instances, such as Dong et al. (2017,2018). Hence, these studies are worth citing here to show that the issue of atmospheric loss around low-mass stars is subject to ambiguities:

Dong, C. et al. (2017). Is Proxima Centauri b Habitable? A Study of Atmospheric Loss. Astrophys. J. Lett., 837, L26

Dong, C. et al. (2018). Atmospheric escape from the TRAPPIST-1 planets and implications for habitability. Proc. Natl. Acad. Sci. USA, 115, 260

Author Response

I thank the reviewer for their suggestions on the draft, and find that incorporating them helps to improve the paper.

This is an interesting and comprehensive work that examines how the multiverse hypothesis may be falsified through the study of biosignatures in the future. I believe that the manuscript merits publication in the journal, but I would like to see a few points explained better.

(a) On pg. 4, the ansatz considered is proportional to the stellar lifetime. However, over a given fraction of the stellar lifetime, it is possible that habitability has a non-linear dependence on the age. For example, the species richness (no. of species) might be very roughly modelled as an exponential up to some cutoff time as suggested in Lingam & Loeb (2017), which can be cited here. Perhaps the author can comment on what will happen in case an exponential dependence on the total stellar lifetime (or a constant fraction of it) is introduced.

Lingam, M. and Loeb, A. (2017). Reduced Diversity of Life around Proxima Centauri and TRAPPIST-1. Astrophys. J. Lett., 846, L21

In fact, this form is considered on line 287 (though weighted by entropy production).  A pointer to this discussion, as well as an inclusion of this reference, is now included on line 130.

(b) On pg. 5, the author assumes that the fraction of stars with planets of similar size (and presumably temperature) is not dependent on stellar mass. This is a fairly good assumption, and I'd recommend citing Table 1 of Kaltenegger (2017) here.

Kaltenegger, L. (2017). How to characterize habitable worlds and signs of life. Annu Rev. Astron. Astrophys., 55:433

I thank the reviewer for this reference, and have included it in the discussion on line 153.

(c) On pg. 6, just below Eq. (4), a simple physical calculation along the lines of Weisskopf's analysis demonstrates that H_clay is inversely proportional to g. However, the density of all terrestrial planets is not quite the same, owing to which g does not scale linearly with R. Instead, it scales roughly as R^(1.7) because M is proportional to R^(3.7) as shown in the simulations by Zeng et al. (2016). Hence, this correction should be noted and the text altered accordingly.

Zeng, L., Sasselov, D. D., and Jacobsen, S. B. (2016). Mass–radius relation for rocky planets based on PREM. Astrophys. J., 819, 127

Unfortunately, I don’t think I can do this self consistently.  To take the compressibility of planets into account, we need to know why M~R^3.7.  An explanation for the exponent can be found if we take the differential equations (1-3) of

Valencia, D., O'Connell, R. J., & Sasselov, D. (2006). Internal structure of massive terrestrial planets. Icarus, 181(2), 545-554.

And, for dimensional analysis purposes, replace d/dr -> 1/R.  We find

M(R) ~ rho_0 R^3 f ( G rho_0^2 R^2 / K_0 ),

where K_0 is the bulk modulus, and f(t) is a function that interpolates between 1 for small arguments and t^3 for large arguments.  With this, the scaling exponent (defined as d log M/d log R) is between 3 and 9, the particular value set by the fact that compression just starts to become important for Earthlike planets.  However, the reason this scaling does not appear in the final calculation is that the argument of the function f is just a number, independent of physical constants, when we take the planet radius to be set by the retention of only heavy gases!  This is not as mysterious as it seems, since this relies on balancing gravitational and molecular energies, the same condition which dictates when rocks start to compact.  However, this effectively sets the relation to be M(R) ~ const * R^3 in terms of the dependence on constants for terrestrial planets.  In the future I plan to relax the terrestrial assumption, and then I will certainly take this effect into account as well, but for the time being I’m afraid my analysis prevents it.

(d) Eq. (8) is accurate because the temperature of the planet is much smaller than that of its host star. Thermodynamic efficiency a la Carnot is also close to 100% and enables the effective utilization of photons. However, when it comes to planets around cool brown dwarfs, one cannot automatically assume that T_planet is much lower than T_BD. Of course, the author need not mention this aspect since they are dealing with stars.

I’ve added a short note making this assumption explicit on line 203.

(e) In principle, there do not seem to be strong a priori reasons, from the perspective of evolving photosystems (with reaction centres and light-harvesting complexes), that prevent the use of wavelengths<400 nm="" and="">1100 nm, especially the latter. Multi-photon capture schemes are theoretically feasible, as outlined in Wolstencroft & Raven (2002) and subsequent papers. It may be worth mentioning this cautionary statement on pg. 8.

Wolstencroft, R. D. and Raven, J. A. (2002). Photosynthesis: Likelihood of Occurrence and Possibility of Detection on Earth-like Planets. Icarus, 157:535

I discuss these values in much more depth in the first paper of this series (published in the same journal issue): the former comes from photodissociation bounds, and the latter from the thermal background, though in the discussion I note that these are not necessarily known, and include some references that have an enlarged range (including the one the reviewer mentions).  For the present discussion, I included a sentence on line 230 briefly explaining the fiducial values chosen, refer to the full discussion in ref [1], and note on line 233 how the results do not depend sensitively on the range used.

(f) In pgs. 11 and 12, the importance of the weathering rate is rightly pointed out as it provides the supply of nutrients. However, the steady-state concentration depends not only on the influx of nutrients but also on their characteristic residence times in the environments under question. For example, in the oceans, this is constrained by the hydrothermal circulation time, which in turn depends on both the degree of internal heat as well as a number of geochemical factors. As it is virtually impossible to model the underlying complexities of nutrient concentrations, I suggest that the author should briefly note that nutrient turnover timescales and their accompanying phenomena are not taken into account herein.

I’ve added this to the discussion after eqn (14).  It would be interesting to undertake this project in the future, but I agree with the reviewer that it feels virtually impossible at the present moment.

(g) With regards to plate tectonics on pg. 14, the most recent numerical models indicate that stagnant-lid tectonics can support "habitable" conditions for Gyr timescales, and that plate tectonics may not be necessary for the stabilization of climate. See, for example, Foley (2019) and references therein:

Foley, B. J. (2019). Habitability of Earth-like stagnant lid planets: Climate evolution and recovery from snowball states. arXiv:1903.12111

I’ve now included this reference on line 336, and think it adds to a more balanced discussion.

(h) The discussion at the beginning of pg. 18 was interesting, and I'd appreciate it if the author could clear up a certain point for me. Suppose, for the sake of argument, that one subscribes to a "strong" version of evolutionary convergence wherein the PDF for the number of hard steps is close to a delta function centered on the no. of hard steps required on Earth. What would this imply for the likelihood of intelligence to emerge on an average planet?

The discussion starting on line 476 is perhaps not as clear as it should have been, and so I have updated it.  If the strong convergence viewpoint is adopted, then the number of hard steps will always be the same, and so we expect to have emerged on an average planet.  The trouble with the hard step model comes when one views hardness as a continuum, which treats eukaryogenesis on the same footing as the evolution of teeth, etc, just at different points on the distribution.  In this view, we would expect to have been born on a planet where eukaryogenesis was accidentally facilitated by random environmental factors (as some argue the Huronian glaciation did), rather than a planet where life brute forced a solution under typical circumstances.

(i) On pg. 21, subsequent numerical simulations after Airapetian et al. (2017) [Ref. 62] have shown that the retention of atmospheres on Gyr timescales might be possible in some instances, such as Dong et al. (2017,2018). Hence, these studies are worth citing here to show that the issue of atmospheric loss around low-mass stars is subject to ambiguities:

Dong, C. et al. (2017). Is Proxima Centauri b Habitable? A Study of Atmospheric Loss. Astrophys. J. Lett., 837, L26

Dong, C. et al. (2018). Atmospheric escape from the TRAPPIST-1 planets and implications for habitability. Proc. Natl. Acad. Sci. USA, 115, 260

I’ve added these references to the discussion after eqn (32).

Reviewer 2 Report

The aim of the paper is to determine the probability of developing life in a universe as a function of fundamental constants (fine structure constant, ratio of electron to proton mass and ratio of proton to Planck mass). Paper provides a comprehensive discussion of the habitability criterion formulated by the author. Although hypothetical, yet the value of the paper is in proposing a certain approach to discuss habitability criteria formulated in terms of fundamental constants.

I recommend the paper to be published in the Universe.

Author Response

N/A